# Polar Operators for Structured Sparse Estimation

**Xinhua Zhang**
Machine Learning Research Group
National ICT Australia and ANU
xinhua.zhang@anu.edu.au

**Yaoliang Yu  and  Dale Schuurmans**
Department of Computing Science, University of Alberta
Edmonton, Alberta T6G 2E8, Canada
{yaoliang,dale}@cs.ualberta.ca

## Abstract

Structured sparse estimation has become an important technique in many areas of data analysis. Unfortunately, these estimators normally create computational difficulties that entail sophisticated algorithms. Our first contribution is to uncover a rich class of structured sparse regularizers whose *polar operator* can be evaluated efficiently. With such an operator, a simple conditional gradient method can then be developed that, when combined with smoothing and local optimization, significantly reduces training time vs. the state of the art. We also demonstrate a new reduction of polar to proximal maps that enables more efficient latent fused lasso.

## 1   Introduction

Sparsity is an important concept in high-dimensional statistics [1] and signal processing [2] that has led to important application successes by reducing model complexity and improving interpretability of the results. Standard computational strategies such as greedy feature selection [3] and generic convex optimization [4–7] can be used to implement simple sparse estimators. However, sophisticated notions of *structured* sparsity have been recently developed that can encode *combinatorial* patterns over variable subsets [8]. Although combinatorial structure greatly enhances modeling capability, it also creates computational challenges that require sophisticated optimization approaches. For example, current structured sparse estimators often adopt an *accelerated proximal gradient* (APG) strategy [9, 10], which has a low per-step complexity and enjoys an optimal convergence rate among black-box first-order procedures [10]. Unfortunately, APG must also compute a *proximal update* (PU) of the nonsmooth regularizer during each iteration. Not only does the PU require a highly nontrivial computation for structured regularizers [4]—*e.g.*, requiring tailored network flow algorithms in existing cases [5, 11, 12]—it yields dense intermediate iterates. Recently, [6] has demonstrated a class of regularizers where the corresponding PUs can be computed by a sequence of submodular function minimizations, but such an approach remains expensive.

Instead, in this paper, we demonstrate that an alternative approach can be more effective for many structured regularizers. We base our development on the *generalized conditional gradient* (GCG) algorithm [13, 14], which also demonstrates promise for sparse model optimization. Although GCG possesses a slower convergence rate than APG, it demonstrates competitive performance if its updates are interleaved with local optimization [14–16]. Moreover, GCG produces sparse intermediate iterates, which allows additional sparsity control. Importantly, unlike APG, GCG requires computing the *polar* of the regularizer, instead of the PU, in each step. This difference allows important new approaches for characterizing and evaluating structured sparse regularizers.

Our first main contribution is to characterize a rich class of structured sparse regularizers that allow efficient computation of their polar operator. In particular, motivated by [6], we consider a family of structured sparse regularizers induced by a cost function on variable subsets. By introducing a "lifting" construction, we show how these regularizers can be expressed as linear functions, which after some reformulation, allows efficient evaluation by a simple linear program (LP). Important examples covered include overlapping group lasso [5] and path regularization in directed acyclic graphs [12]. By exploiting additional structure in these cases, the LP can be reduced to a piecewise

linear objective over a simple domain, allowing further reduction in computation time via smoothing [17]. For example, for the overlapping group lasso with $n$ groups where each variable belongs to at most $r$ groups, the cost of evaluating the polar operator can be reduced from $O(rn^3)$ to $O(rn\sqrt{n}/\epsilon)$ for a desired accuracy of $\epsilon$. Encouraged by the superior performance of GCG in these cases, we then provide a simple reduction of the polar operator to the PU. This reduction makes it possible to extend GCG to cases where the PU is easy to compute. To illustrate the usefulness of this reduction we provide an efficient new algorithm for solving the fused latent lasso [18].

## 2  Structured Sparse Models

Consider the standard regularized risk minimization framework

$$\min_{\mathbf{w}\in\mathbb{R}^n} f(\mathbf{w}) + \lambda\,\Omega(\mathbf{w}), \tag{1}$$

where $f$ is the empirical risk, assumed to be convex with a Lipschitz continuous gradient, and $\Omega$ is a convex, positively homogeneous regularizer, *i.e.* a *gauge* [19, §4]. Let $2^{[n]}$ denote the power set of $[n] := \{1, \ldots, n\}$, and let $\overline{\mathbb{R}}_+ := \mathbb{R}_+ \cup \{\infty\}$. Recently, [6] has established a principled method for deriving regularizers from a subset cost function $F : 2^{[n]} \to \overline{\mathbb{R}}_+$ based on defining the gauge:

$$\Omega_F(\mathbf{w}) = \inf\{\gamma \geq 0 : \mathbf{w} \in \gamma \operatorname{conv}(S_F)\}, \text{ where } S_F = \left\{\mathbf{w}_A : \|\mathbf{w}_A\|_{\tilde{p}}^p = 1/F(A), \emptyset \neq A \subseteq [n]\right\}. \tag{2}$$

Here $\gamma$ is a scalar, $\operatorname{conv}(S_F)$ denotes the convex hull of the set $S_F$, $\tilde{p}, p \geq 1$ with $\frac{1}{\tilde{p}} + \frac{1}{p} = 1$, $\|\cdot\|_p$ throughout is the usual $\ell_p$-norm, and $\mathbf{w}_A$ denotes a duplicate of $\mathbf{w}$ with all coordinates not in $A$ set to 0. Note that we have tacitly assumed $F(A) = 0$ iff $A = \emptyset$ in (2). The gauge $\Omega_F$ defined in (2) is also known as the *atomic norm* with the set of atoms $S_F$ [20]. It will be useful to recall that the *polar* of a gauge $\Omega$ is defined by [19, §15]:

$$\Omega^\circ(\mathbf{g}) := \sup_{\mathbf{w}}\{\langle \mathbf{g}, \mathbf{w}\rangle : \Omega(\mathbf{w}) \leq 1\}. \tag{3}$$

In particular, the polar of a norm is its dual norm. (Recall that any norm is also a gauge.) For the specific gauge $\Omega_F$ defined in (2), its polar is simply the support function of $S_F$ [19, Theorem 13.2]:

$$\Omega_F^\circ(\mathbf{g}) = \max_{\mathbf{w}\in S_F} \langle \mathbf{g}, \mathbf{w}\rangle = \max_{\emptyset \neq A \subseteq [n]} \|\mathbf{g}_A\|_p / [F(A)]^{1/p}. \tag{4}$$

(The first equality uses the definition of support function, and the second follows from (2).) By varying $\tilde{p}$ and $F$, one can generate a class of sparsity inducing regularizers that includes most current proposals [6]. For instance, if $F(A) = 1$ whenever $|A|$ (the cardinality of $A$) is 1, and $F(A) = \infty$ for $|A| > 1$, then $\Omega_F^\circ$ is the $\ell_\infty$ norm and $\Omega_F$ is the usual $\ell_1$ norm. More importantly, one can encode structural information through the cost function $F$, which selects and establishes preferences over the set of atoms $S_F$. As pointed out in [6], when $F$ is submodular, (4) can be evaluated by a secant method with submodular minimizations ([21, §8.4], see also Appendix B). However, as we will show, it is possible to do significantly better by completely avoiding submodular optimization. Before presenting our main results, we first review the state of the art for solving (1), and demonstrate how the performance of current methods can hinge on efficient computation of (4).

### 2.1  Optimization Algorithms

A standard approach for minimizing (1) is the *accelerated proximal gradient* (APG) algorithm [9, 10], where each iteration involves solving the *proximal update* (PU): $\mathbf{w}_{k+1} = \arg\min_{\mathbf{w}} \langle \mathbf{d}_k, \mathbf{w}\rangle + \frac{1}{2s_k}\|\mathbf{w} - \mathbf{w}_k\|_2^2 + \lambda\Omega_F(\mathbf{w})$, for some step size $s_k$ and descent direction $\mathbf{d}_k$. Although it can be shown that APG finds an $\epsilon$ accurate solution in $O(1/\sqrt{\epsilon})$ iterations [9, 10], each update can be quite difficult to compute when $\Omega_F$ encodes combinatorial structure, as noted in the introduction.

An alternative approach to solving (1) is the *generalized conditional gradient* (GCG) method [13, 14], which has recently received renewed attention. Unlike APG, GCG only requires the *polar operator* of the regularizer $\Omega_F$ to be computed in each iteration, given by the argument of (4):

$$\mathbb{P}_F^\circ(\mathbf{g}) = \arg\max_{\mathbf{w}\in S_F} \langle \mathbf{g}, \mathbf{w}\rangle = F(C)^{\frac{-1}{p}} \arg\max_{\mathbf{w}:\|\mathbf{w}\|_{\tilde{p}}=1} \langle \mathbf{g}_C, \mathbf{w}\rangle \text{ for } C = \arg\max_{\emptyset \neq A \subseteq [n]} \|\mathbf{g}_A\|_p^p / F(A). \tag{5}$$

Algorithm 1 outlines a GCG procedure for solving (1) that only requires the evaluation of $\mathbb{P}_F^\circ$ in each iteration without needing the full PU to be computed. The algorithm is quite simple: Line 3

**Algorithm 1** Generalized conditional gradient (GCG) for optimizing (1).

1: Initialize $\mathbf{w}_0 \leftarrow \mathbf{0}, \ s_0 \leftarrow 0, \ \boldsymbol{\ell}_0 \leftarrow \mathbf{0}$.
2: **for** $k = 0, 1, \dots$ **do**
3:      Polar operator: $\mathbf{v}_k \leftarrow \mathbb{P}_F^\circ(\mathbf{g}_k), \ A_k \leftarrow C(\mathbf{g}_k)$, where $\mathbf{g}_k = -\nabla f(\mathbf{w}_k)$ and $C$ is defined in (5).
4:      2-D Conic search: $(\alpha, \beta) := \arg\min_{\alpha \geq 0, \beta \geq 0} f(\alpha \mathbf{w}_k + \beta \mathbf{v}_k) + \lambda(\alpha s_k + \beta)$.
5:      Local re-optimization: $\{\mathbf{u}^i\}_1^k := \arg\min_{\{\mathbf{u}^i = \mathbf{u}_{A_i}^i\}} f(\sum_i \mathbf{u}^i) + \lambda \sum_i F(A_i)^{\frac{1}{p}} \|\mathbf{u}^i\|_{\tilde{p}}$
               where the $\{\mathbf{u}^i\}$ are initialized by $\mathbf{u}^i = \alpha \boldsymbol{\ell}_i$ for $i < k$ and $\mathbf{u}^i = \beta \mathbf{v}^i$ for $i = k$.
6:      $\mathbf{w}_{k+1} \leftarrow \sum_i \mathbf{u}^i, \ \boldsymbol{\ell}_i \leftarrow \mathbf{u}^i$ for $i \leq k, \ s_{k+1} \leftarrow \sum_i F(A_i)^{\frac{1}{p}} \|\mathbf{u}^i\|_{\tilde{p}}$.
7: **end for**

evaluates the polar operator, which provides a descent direction $\mathbf{v}_k$; Line 4 finds the optimal step sizes for combining the current iterate $\mathbf{w}_k$ with the direction $\mathbf{v}_k$; and Line 5 locally improves the objective (1) by maintaining the same support patterns but re-optimizing the parameters. It has been shown that GCG can find an $\epsilon$ accurate solution to (1) in $O(1/\epsilon)$ steps, provided only that the polar (5) is computed to $\epsilon$ accuracy [14]. Although GCG has a slower theoretical convergence rate than APG, the introduction of local optimization (Line 5) often yields faster convergence in practice [14–16]. Importantly, Line 5 does not increase the sparsity of the intermediate iterates. Our main goal in this paper therefore is to extend this GCG approach to structured sparse models by developing efficient algorithms for computing the polar operator for the structured regularizers defined in (2).

## 3 Polar Operators for Atomic Norms

Let $\mathbf{1}$ denote the vector of all 1s with length determined by context. Our first main contribution is to develop a general class of atomic norm regularizers whose polar operator (5) can be computed efficiently. To begin, consider the case of a (partially) *linear* function $F$ where there exists a $\mathbf{c} \in \mathbb{R}^n$ such that $F(A) = \langle \mathbf{c}, \mathbf{1}_A \rangle$ for all $A \in \operatorname{dom} F$ (note that the domain need not be a lattice). A few useful regularizers can be generated by linear functions: for example, the $\ell_1$ norm can be derived from $F(A) = \langle \mathbf{1}, \mathbf{1}_A \rangle$ for $|A| = 1$, which is linear. Unfortunately, linearity is too restrictive to capture most structured regularizers of interest, therefore we will need to expand the space of functions $F$ we consider. To do so, we introduce the more general class of *marginalized linear functions*: we say that $F$ is marginalized linear if there exists a nonnegative linear function $M$ on an extended domain $2^{[n+l]}$ such that its marginalization to $2^{[n]}$ is exactly $F$:

$$F(A) = \min_{B:A \subseteq B \subseteq [n+l]} M(B), \quad \forall A \subseteq [n]. \tag{6}$$

Essentially, such a function $F$ is "lifted" to a larger domain where it becomes linear. The key question is whether the polar $\Omega_F^\circ$ can be efficiently evaluated for such functions.

To develop an efficient procedure for computing the polar $\Omega_F^\circ$, first consider the simpler case of computing the polar $\Omega_M^\circ$ for a nonnegative linear function $M$. Note that by linearity the function $M$ can be expressed as $M(B) = \langle \mathbf{b}, \mathbf{1}_B \rangle$ for $B \in \operatorname{dom} M \subseteq 2^{[n+l]}$ ($\mathbf{b} \in \mathbb{R}_+^{n+l}$). Since the effective domain of $M$ need not be the whole space in general, we make use of the specialized polytope:

$$P := \operatorname{conv}\{\mathbf{1}_B : B \in \operatorname{dom} M\} \subseteq [0,1]^{n+l}. \tag{7}$$

Note $P$ may have exponentially many faces. From the definition (4) one can then re-express the polar $\Omega_M^\circ$ as:

$$\Omega_M^\circ(\mathbf{g}) = \max_{\emptyset \neq B \in \operatorname{dom} M} \|\mathbf{g}_B\|_p / M(B)^{1/p} = \left( \max_{\mathbf{0} \neq \mathbf{w} \in P} \frac{\langle \tilde{\mathbf{g}}, \mathbf{w} \rangle}{\langle \mathbf{b}, \mathbf{w} \rangle} \right)^{1/p} \quad \text{where } \tilde{g}_i = |g_i|^p \ \forall i, \tag{8}$$

where we have used the fact that the linear-fractional objective must attain its maximum at vertices of $P$; that is, at $\mathbf{1}_B$ for some $B \in \operatorname{dom} M$. Although the linear-fractional program (8) can be reduced to a sequence of LPs using the classical method of [22], a single LP suffices for our purposes. Indeed, let us first remove the constraint $\mathbf{w} \neq \mathbf{0}$ by considering the alternative polytope:

$$Q := P \cap \{\mathbf{w} \in \mathbb{R}^{n+l} : \langle \mathbf{1}, \mathbf{w} \rangle \geq 1\}. \tag{9}$$

As shown in Appendix A, all vertices of $Q$ are scalar multiples of the nonzero vertices of $P$. Since the objective in (8) is scale invariant, we can restrict the constraints to $\mathbf{w} \in Q$. Then, by applying transformations $\tilde{\mathbf{w}} = \mathbf{w} / \langle \mathbf{b}, \mathbf{w} \rangle, \sigma = 1 / \langle \mathbf{b}, \mathbf{w} \rangle$, problem (8) can be equivalently re-expressed by:

$$\max_{\tilde{\mathbf{w}}, \sigma > 0} \ \langle \tilde{\mathbf{g}}, \tilde{\mathbf{w}} \rangle, \text{ subject to } \tilde{\mathbf{w}} \in \sigma Q, \ \langle \mathbf{b}, \tilde{\mathbf{w}} \rangle = 1. \tag{10}$$

Of course, whether this LP can be solved efficiently depends on the structure of $Q$ (and of $P$ indeed).

Finally, we note that the same formulation allows the polar to be efficiently computed for a *marginalized* linear function $F$ via a simple reduction: Consider any $\mathbf{g} \in \mathbb{R}^n$ and let $[\mathbf{g}; \mathbf{0}] \in \mathbb{R}^{n+l}$ denote $\mathbf{g}$ padded by $l$ zeros. Then $\Omega_F^\circ(\mathbf{g}) = \Omega_M^\circ([\mathbf{g}; \mathbf{0}])$ for all $\mathbf{g} \in \mathbb{R}^n$ because

$$\max_{\emptyset \neq A \subseteq [n]} \frac{\|\mathbf{g}_A\|_p^p}{F(A)} = \max_{\emptyset \neq A \subseteq [n]} \frac{\|\mathbf{g}_A\|_p^p}{\min_{B : A \subseteq B \subseteq [n+l]} M(B)} = \max_{\emptyset \neq A \subseteq B} \frac{\|\mathbf{g}_A\|_p^p}{M(B)} = \max_{B : \emptyset \neq B \subseteq [n+l]} \frac{\|[\mathbf{g}; \mathbf{0}]_B\|_p^p}{M(B)}. \quad (11)$$

To see the last equality, fixing $B$ the optimal $A$ is attained at $A = B \cap [n]$. If $B \cap [n]$ is empty, then $\|[\mathbf{g}; \mathbf{0}]_B\| = 0$ and the corresponding $B$ cannot be the maximizer of the last term, unless $\Omega_F^\circ(\mathbf{g}) = 0$ in which case it is easy to see $\Omega_M^\circ([\mathbf{g}; \mathbf{0}]) = 0$.

Although we have kept our development general so far, the idea is clear: once an appropriate "lifting" has been found so that the polytope $Q$ in (9) can be compactly represented, the polar (5) can be reformulated as the LP (10), for which efficient implementations can be sought. We now demonstrate this new methodology for the two important structured regularizers: group sparsity and path coding.

## 3.1 Group Sparsity

For a general formulation of group sparsity, let $\mathcal{G} \subseteq 2^{[n]}$ be a set of variable groups (subsets) that possibly overlap [3, 6, 7]. Here we use $i \in [n]$ to index variables and $G \in \mathcal{G}$ to index groups. Consider the cost function over variable groups $F_{\mathbf{g}} : 2^{[n]} \to \mathbb{R}_+$ defined by:

$$F_{\mathbf{g}}(A) = \sum_{G \in \mathcal{G}} c_G \, \mathbb{I}(A \cap G \neq \emptyset), \quad (12)$$

where $c_G$ is a nonnegative cost and $\mathbb{I}$ is an indicator such that $\mathbb{I}(\cdot) = 1$ if its argument is true, and $0$ otherwise. The value $F_{\mathbf{g}}(A)$ provides a weighted count of how many groups overlap with $A$. Unfortunately, $F_{\mathbf{g}}$ is not linear, so we need to re-express it to recover an efficient polar operator. To do so, augment the domain by adding $l = |\mathcal{G}|$ variables such that each new variable $G$ corresponds to a group $G$. Then define a weight vector $\mathbf{b} \in \mathbb{R}_+^{n+l}$ such that $b_i = 0$ for $i \leq n$ and $b_G = c_G$ for $n < G \leq n + l$. Finally, consider the linear cost function $M_{\mathbf{g}} : 2^{[n+l]} \to \overline{\mathbb{R}}_+$ defined by:

$$M_{\mathbf{g}}(B) = \langle \mathbf{b}, \mathbf{1}_B \rangle \text{ if } i \in B \Rightarrow G \in B, \, \forall \, i \in G \in \mathcal{G}; \quad M_{\mathbf{g}}(B) = \infty \text{ otherwise}. \quad (13)$$

The constraint ensures that if a variable $i \leq n$ appears in the set $B$, then every variable $G$ corresponding to a group $G$ that contains $i$ must also appear in $B$. By construction, $M_{\mathbf{g}}$ is a nonnegative linear function. It is also easy to verify that $F_{\mathbf{g}}$ satisfies (6) with respect to $M_{\mathbf{g}}$.

To compute the corresponding polar, observe that the effective domain of $M_{\mathbf{g}}$ is a lattice, hence (4) can be solved by combinatorial methods. However, we can do better by exploiting problem structure in the LP. For example, observe that the polytope (7) can now be compactly represented as:

$$P_{\mathbf{g}} = \{\mathbf{w} \in \mathbb{R}^{n+l} : \mathbf{0} \leq \mathbf{w} \leq \mathbf{1}, w_i \leq w_G, \forall \, i \in G \in \mathcal{G}\}. \quad (14)$$

Indeed, it is easy to verify that the integral vectors in $P_{\mathbf{g}}$ are precisely $\{\mathbf{1}_B : B \in \mathrm{dom}\, M_{\mathbf{g}}\}$. Moreover, the linear constraint in (14) is totally unimodular (TUM) since it is the incidence matrix of a bipartite graph (variables and groups), hence $P_{\mathbf{g}}$ is the convex hull of its integral vectors [23]. Using the fact that the scalar $\sigma$ in (10) admits a closed form solution $\sigma = \langle \mathbf{1}, \tilde{\mathbf{w}} \rangle$ in this case, the LP (10) can be reduced to:

$$\max_{\tilde{\mathbf{w}}} \sum_{i \in [n]} \tilde{g}_i \min_{G : i \in G \in \mathcal{G}} \tilde{w}_G, \text{ subject to } \tilde{\mathbf{w}} \geq 0, \sum_{G \in \mathcal{G}} b_G \tilde{w}_G = 1. \quad (15)$$

Note only $\{\tilde{w}_G\}$ appear in the problem as implicitly $\tilde{w}_i = \min_{G : i \in G} \tilde{w}_G, \, \forall \, i \in [n]$. This is now just a piecewise linear objective over a (reweighted) simplex. Since projecting to a simplex can be performed in linear time, the smoothing method of [17] can be used to obtain a very efficient implementation. We illustrate a particular case where each variable $i \in [n]$ belongs to at most $r > 1$ groups. (Appendix D considers when the groups form a directed acyclic graph.)

**Proposition 1** *Let $h(\tilde{\mathbf{w}})$ denote the* negated *objective of* (15)*. Then for any $\epsilon > 0$, $h_\epsilon(\tilde{\mathbf{w}}) := \frac{\epsilon}{n \log r} \sum_{i \in [n]} \log \sum_{G : i \in G} r^{-n \tilde{g}_i \tilde{w}_G / \epsilon}$ satisfies: (i) the gradient of $h_\epsilon$ is $\left(\frac{n}{\epsilon} \|\tilde{\mathbf{g}}\|_\infty^2 \log r\right)$-Lipschitz, (ii) $h(\tilde{\mathbf{w}}) - h_\epsilon(\tilde{\mathbf{w}}) \in (-\epsilon, 0]$ for all $\tilde{\mathbf{w}}$, and (iii) the gradient of $h_\epsilon$ can be computed in $O(nr)$ time.*

(The proof is given in Appendix C.) With this construction, APG can be run on $h_\epsilon$ to achieve a $2\epsilon$ accurate solution to (15) within $O(\frac{1}{\epsilon}\sqrt{n \log r})$ steps [17], using a total time cost of $O(\frac{nr}{\epsilon}\sqrt{n \log r})$. Note that this is significantly cheaper than the $O(n^2(l+n)r)$ worst case complexity of [11, Algorithm 2]. More importantly, we gain explicit control of the trade-off between accuracy $\epsilon$ and computational cost. A detailed comparison to related approaches is given in Appendix B.1 and E.

## 3.2 Path Coding

Another interesting regularizer, recently investigated by [12], is determined by path costs in a directed acyclic graph (DAG) defined over the set of variables $i \in [n]$. For convenience, we add two nodes, a source $s$ and a sink $t$, with dummy edges $(s, i)$ and $(i, t)$ for all $i \in [n]$. An $(s, t)$-path (or simply path) is then given by a sequence $(s, i_1), (i_1, i_2), \ldots, (i_{k-1}, i_k), (i_k, t)$ with $k \geq 1$. A non-negative cost is associated with each edge including $(s, i)$ and $(i, t)$, so the cost of a path is the sum of its edge costs. A regularizer can then be defined by (2) applied to the cost function $F_{\mathsf{p}} : 2^{[n]} \to \overline{\mathbb{R}}_+$

$$F_{\mathsf{p}}(A) = \begin{cases} \text{cost of the path} & \text{if the nodes in } A \text{ form an } (s, t)\text{-path (unique for DAG)} \\ \infty & \text{if such a path does not exist} \end{cases}. \quad (16)$$

Note $F_{\mathsf{p}}$ is *not* submodular. Although $F_{\mathsf{p}}$ is not linear, a similar "lifting" construction can be used to show that it is marginalized linear, hence it supports efficient computation of the polar. To explain the construction, let $V := [n] \cup \{s, t\}$ be the node set including $s$ and $t$, $E$ be the edge set including $(s, i)$ and $(i, t)$, $T = V \cup E$, and let $\mathbf{b} \in \mathbb{R}_+^{|T|}$ be the concatenation of zeros for node costs and the given edge costs. Let $m := |E|$ be the number of edges. It is then easy to verify that $F_{\mathsf{p}}$ satisfies (6) with respect to the linear cost function $M_{\mathsf{p}} : 2^T \to \overline{\mathbb{R}}_+$ defined by:

$$M_{\mathsf{p}}(B) = \langle \mathbf{b}, \mathbf{1}_B \rangle \text{ if } B \text{ represents a path}; \quad \infty \text{ otherwise.} \quad (17)$$

To efficiently compute the resulting polar, we consider the form (8) using $\tilde{g}_i = |g_i|^p \; \forall i$ as before:

$$\Omega_{M_{\mathsf{p}}}^\circ(\mathbf{g}) = \max_{\mathbf{0} \neq \mathbf{w} \in [0,1]^{|T|}} \frac{\langle \tilde{\mathbf{g}}, \mathbf{w} \rangle}{\langle \mathbf{b}, \mathbf{w} \rangle}, \quad \text{s.t.} \quad w_i = \sum_{j:(i,j)\in E} w_{ij} = \sum_{k:(k,i)\in E} w_{ki}, \; \forall i \in [n]. \quad (18)$$

Here the constraints form the well-known flow polytope whose vertices are exactly all the paths in a DAG. Similar to (15), the normalized LP (10) can be simplified by solving for the scalar $\sigma$ to obtain:

$$\max_{\tilde{\mathbf{w}} \geq \mathbf{0}} \sum_{i\in[n]} \tilde{g}_i \left( \sum_{j:(i,j)\in E} \tilde{w}_{ij} + \sum_{k:(k,i)\in E} \tilde{w}_{ki} \right), \text{ s.t. } \langle \mathbf{b}, \tilde{\mathbf{w}} \rangle = 1, \sum_{j:(i,j)\in E} \tilde{w}_{ij} = \sum_{k:(k,i)\in E} \tilde{w}_{ki}, \forall i \in [n]. \quad (19)$$

Due to the extra constraints, the LP (19) is more complicated than (15) obtained for group sparsity. Nevertheless, after some reformulation (essentially dualization), (19) can still be converted to a simple piecewise linear objective, hence it is amenable to smoothing; see Appendix F for details. To find a $2\epsilon$ accurate solution, the cutting plane method takes $O(\frac{mn}{\epsilon^2})$ computations to optimize the nonsmooth piecewise linear objective, while APG needs $O(\frac{1}{\epsilon}\sqrt{n})$ steps to optimize the smoothed objective, using a total time cost of $O(\frac{m}{\epsilon}\sqrt{n})$. This too is faster than the $O(nm)$ worst case complexity of [12, Appendix D.5] in the regime where $n$ is large and the desired accuracy $\epsilon$ is moderate.

## 4 Generalizing Beyond Atomic Norms

Although we find the above approach to be effective, many useful regularizers are not expressed in form of an atomic norm (2), which makes evaluation of the polar a challenge and thus creates difficulty in applying Algorithm 1. For example, another important class of structured sparse regularizers is given by an alternative, composite gauge construction:

$$\Omega_s(\mathbf{w}) = \sum_i \kappa_i(\mathbf{w}), \text{ where } \kappa_i \text{ is a } closed \text{ gauge that can be different for different } i. \quad (20)$$

The polar for such a regularizer is given by $\Omega_s^\circ(\mathbf{g}) = \inf\{\max_i \kappa_i^\circ(\mathbf{w}^i) : \sum_i \mathbf{w}^i = \mathbf{g}\}$, where each $\mathbf{w}^i$ is an independent vector and $\kappa_i^\circ$ corresponds to the polar of $\kappa_i$ (proof given in Appendix H). Unfortunately, a polar in this form does not appear to be easy to compute. However, for some regularizers in the form (20) the following proximal objective can indeed be computed efficiently:

$$\mathsf{Prox}_\Omega(\mathbf{g}) = \min_{\boldsymbol{\theta}} \tfrac{1}{2}\|\mathbf{g} - \boldsymbol{\theta}\|_2^2 + \Omega(\boldsymbol{\theta}), \qquad \mathsf{ArgProx}_\Omega(\mathbf{g}) = \arg\min_{\boldsymbol{\theta}} \tfrac{1}{2}\|\mathbf{g} - \boldsymbol{\theta}\|_2^2 + \Omega(\boldsymbol{\theta}). \quad (21)$$

The key observation is that computing $\Omega^\circ$ can be efficiently reduced to just computing $\mathsf{Prox}_\Omega$.

**Proposition 2** *For any closed gauge $\Omega$, its polar $\Omega^\circ$ can be equivalently expressed by:*

$$\Omega^\circ(\mathbf{g}) = \inf\{\zeta \geq 0 : \mathsf{Prox}_{\zeta\Omega}(\mathbf{g}) = \tfrac{1}{2}\|\mathbf{g}\|_2^2\}. \quad (22)$$

(The proof is included in Appendix I.) Since the left hand side of the inner constraint is decreasing in $\zeta$, one can efficiently compute the polar $\Omega^\circ$ by a simple root finding search in $\zeta$. Thus, regularizers in the form of (20) can still be accommodated in an efficient GCG method in the form of Algorithm 1.

## 4.1 Latent Fused Lasso

To demonstrate the usefulness of this reduction we consider the recently proposed latent fused lasso model [18], where for given data $X \in \mathbb{R}^{m \times n}$ one seeks a dictionary matrix $W \in \mathbb{R}^{m \times t}$ and coefficient matrix $U \in \mathbb{R}^{t \times n}$ that allow $X$ to be accurately reconstructed from a dictionary that has desired structure. In particular, for a reconstruction loss $f$, the problem is specified by:

$$\min_{W, U \in \mathcal{U}} f(WU, X) + \Omega_p(W), \quad \text{where} \quad \Omega_p(W) = \sum_i \left( \lambda_1 \|W_{:i}\|_p + \lambda_2 \|W_{:i}\|_{\mathsf{TV}} \right), \quad (23)$$

such that $\| \cdot \|_{\mathsf{TV}}$ is given by $\|\mathbf{w}\|_{\mathsf{TV}} = \sum_{j=1}^{m-1} |w_{j+1} - w_j|$ and $\| \cdot \|_p$ is the usual $\ell_p$-norm. The fused lasso [24] corresponds to $p = 1$. Note that $U$ is constrained to be in a compact set $\mathcal{U}$ to avoid degeneracy. To ease notation, we assume w.l.o.g. $\lambda_1 = \lambda_2 = 1$.

The main motivation for this regularizer arises from biostatistics, where one wishes to identify DNA copy number variations *simultaneously* for a group of related samples [18]. In this case the total variation norm $\| \cdot \|_{\mathsf{TV}}$ encourages the dictionary to vary smoothly from entry to entry while the $\ell_p$ norm shrinks the dictionary so that few latent features are selected. Conveniently, $\Omega_p$ decomposes along the columns of $W$, so one can apply the reduction in Proposition 2 to compute its polar assuming $\mathsf{Prox}_{\Omega_p}$ can be efficiently computed. Solving $\mathsf{Prox}_{\Omega_p}$ appears non-trivial due to the composition of two *overlapping* norms, however [25] showed that for $p = 1$ the polar can be solved efficiently by computing $\mathsf{Prox}$ for each of the two norms successively. Here we extend this results by proving in Appendix J that the same fact holds for any $\ell_p$ norm.

**Proposition 3** *For any* $1 \le p \le \infty$, $\mathsf{ArgProx}_{\|\cdot\|_{\mathsf{TV}} + \|\cdot\|_p}(\mathbf{w}) = \mathsf{ArgProx}_{\|\cdot\|_p}\left(\mathsf{ArgProx}_{\|\cdot\|_{\mathsf{TV}}}(\mathbf{w})\right)$.

Since $\mathsf{Prox}_{\|\cdot\|_p}$ is easy to compute, the only remaining problem is to develop an efficient algorithm for computing $\mathsf{Prox}_{\|\cdot\|_{\mathsf{TV}}}$. Although [26] has recently proposed an approximate iterative method, we provide an algorithm in Appendix K that is able to efficiently compute the exact solution. Therefore, by combining this result with Propositions 2 and 3 we are able to efficiently compute the polar $\Omega_p^\circ$ and hence apply Algorithm 1 to solving (23) with respect to $W$.

# 5 Experiments

To investigate the effectiveness of these computational schemes we considered three applications: group lasso, path coding, and latent fused lasso. All algorithms were implemented in Matlab unless otherwise noted.

## 5.1 Group Lasso: CUR-like Matrix Factorization

Our first experiment considered an example of group lasso that is inspired by CUR matrix factorization [27]. Given a data matrix $X \in \mathbb{R}^{n \times d}$, the goal is to compute an approximate factorization $X \approx CUR$, such that $C$ contains a subset of $c$ columns from $X$ and $R$ contains a subset of $r$ rows from $X$. Mairal et al. [11, §5.3] proposed a convex relaxation of this problem:

$$\min_W \tfrac{1}{2} \|X - XWX\|^2 + \lambda \left( \sum_i \|W_{i:}\|_\infty + \sum_j \|W_{:j}\|_\infty \right). \quad (24)$$

Conveniently, the regularizer fits the development of Section 3.1, with $p = 1$ and the groups defined to be the rows and columns of $W$. To evaluate different methods, we used four gene-expression data sets [28]: SRBCT, Brain_Tumor_2, 9_Tumor, and Leukemia2, of sizes $83 \times 2308$, $50 \times 10367$, $60 \times 5762$, and $72 \times 11225$, respectively. The data matrices were first centered columnwise and then rescaled to have unit Frobenius norm.

**Algorithms.** We compared three algorithms: GCG (Algorithm 1) with our polar operator which we call GCG_TUM, GCG with the polar operator of [11, Algorithm 2] (GCG_Secant), and APG (see Section 2.1). The PU in APG uses the routine mexProximalGraph from the SPAMS package [29]. The polar operator of GCG_Secant was implemented with a mex wrapper of a max-flow package [30], while GCG_TUM used L-BFGS to find an optimal solution $\{w_G^*\}$ for the smoothed version of

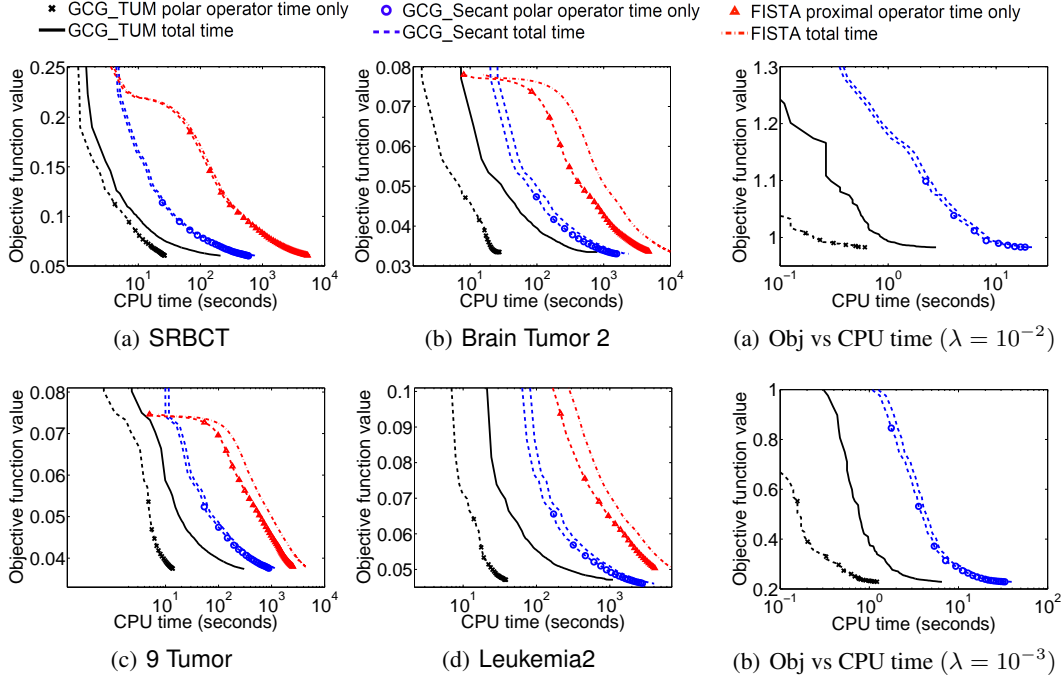

★ GCG_TUM polar operator time only    ○ GCG_Secant polar operator time only    ▲ FISTA proximal operator time only
── GCG_TUM total time    --- GCG_Secant total time    --- FISTA total time

(a) SRBCT      (b) Brain Tumor 2      (a) Obj vs CPU time ($\lambda = 10^{-2}$)

(c) 9 Tumor      (d) Leukemia2      (b) Obj vs CPU time ($\lambda = 10^{-3}$)

Figure 1: Convex CUR matrix factorization results.      Figure 2: Path coding results.

(15) given in Proposition 1, with smoothing parameter $\epsilon$ set to $10^{-3}$. To recover an integral solution it suffices to find an optimal solution to (15) that has the form $w_G = c$ for some groups and $w_G = 0$ for the remainder (such a solution must exist). So we sorted $\{w_G^*\}$ and set the $w_G$ of the smallest $k$ groups to 0, and $w_G$ for the remaining groups set to a common value that satisfies the constraint. The best $k$ can be recovered from $\{0, 1, \ldots, |\mathcal{G}| - 1\}$ in $O(nr)$ time. See more details in Appendix G. Both GCG methods relinquish local optimization (step 5) in Algorithm 1, but use a totally corrective variant of step 4, which allows efficient optimization by L-BFGS-B via pre-computing $X\mathbb{P}_{F_g}^\circ(\mathbf{g}_k)X$.

**Results.** For simplicity, we tested three values for $\lambda$: $10^{-3}$, $10^{-4}$, and $10^{-5}$, which led to increasingly dense solutions. Due to space limitations we only show in Figure 1 the results for $\lambda = 10^{-4}$ which gives moderately sparse solutions. On these data sets, GCG_TUM proves to be an order of magnitude faster than GCG_Secant in computing the polar. As [11] observes, network flow based algorithms often find solutions in practice far more quickly than their theoretical bounds. Thanks to the efficiency of totally corrective update, almost all computations taken by GCG_Secant were devoted to the polar operator. Therefore the acceleration proffered by GCG_TUM in computing the polar leads to a reduction of *overall* optimization time by at least 50%. Finally, APG is always even slower than GCG_Secant by an order of magnitude, with PU taking up the most computation.

## 5.2   Path Coding

Following [12, §4.3], we consider a logistic regression problem where one is given training examples $\mathbf{x}_i \in \mathbb{R}^n$ with corresponding labels $y_i \in \{-1, 1\}$. For this problem, we formulate (1) with a path coding regularizer $\Omega_{F_p}$ and the empirical risk:

$$f(\mathbf{w}) = \sum_i \frac{1}{n_i} \log(1 + \exp(-y_i \langle \mathbf{w}, \mathbf{x}_i \rangle)), \tag{25}$$

where $n_i$ is the number of examples that share the same label as $y_i$. We used the breast cancer data set for this experiment, which consists of 8141 genes and 295 tumors [31]. The gene network is adopted from [32]. Similar to [12, §4.3], we removed all isolated genes (nodes) to which no edge is incident, randomly oriented the raw edges, and removed cycles to form a DAG using the function mexRemoveCyclesGraph in SPAMS. This resulted in 34864 edges and $n = 7910$ nodes.

**Algorithms.** We again considered three methods: APG, GCG with our polar operator (GCG_TUM), and GCG with the polar operator from [12, Algorithm 1], which we label as GCG_Secant. The PU in APG uses the routine mexProximalPathCoding from SPAMS, which solves a quadratic network flow problem. It turns out the time cost for a single call of the PU was enough for GCG_TUM and

GCG_Secant to converge to a final solution, and so the APG result is not included in our plots. We implemented the polar operator for GCG_Secant based on Matlab's built-in shortest path routine graphshortestpath (C++ wrapped by mex). For GCG_TUM, we used cutting plane to solve a variant of the dual of (19) (see Appendix F), which is much simipler than smoothing in implementation, but exhibits similar efficiency in practice. An integral solution can also be naturally recovered in the course of computing the objective. Again, both GCG methods only used totally corrective updates.

**Results.** Figure 2 shows the result for path coding, with the regularization coefficient $\lambda$ set to $10^{-2}$ and $10^{-3}$ so that the solution is moderately sparse. Again it is clear that GCG_TUM is an order of magnitude faster than GCG_Secant.

## 5.3 Latent Fused Lasso

Finally, we compared GCG and APG on the latent fused lasso problem (23). Two algorithms were tested as the PU in APG: our proposed method and the algorithm in [26], which we label as APG-Liu. The synthetic data is generated by following [18]. For each basis (column) of the dictionary, we use the model $\tilde{W}_{ij} = \sum_{s=1}^{S_j} c_s \mathbb{I}(i_s \leq i \leq i_s + l_s)$, where $S_j \in \{3, 5, 8, 10\}$ specifies the number of consecutive blocks in the $j$-th basis, $c_s \in \{\pm 1, \pm 2, \pm 3, \pm 4, \pm 5\}$, $i_s \in \{1, \ldots, m - 10\}$ and $l_s \in \{5, 10, 15, 20\}$, which are the magnitude, starting position, and length of the $s$-th block, respectively. Note that we choose $c_s, i_s, l_s$ randomly (and independently for each block $s$) from their respective sets. The coefficient matrix $\tilde{U}$ are sampled from the Gaussian distribution $N(0, 1)$ (independently for each entry) and normalized to have unit $\ell_2$ norm for each row. Finally, we generate the observation matrix $X = \tilde{W}\tilde{U} + \varepsilon$, with added (zero mean and unit variance) Gaussian noise $\varepsilon$. We set the dimension $m = 300$, the number of samples $n = 200$, and the number of bases (latent dimension) $\tilde{t} = 10$.

Since the noise is Gaussian, we choose the squared loss $f(WU, X) = \frac{1}{2}\|X - WU\|_F^2$, but the algorithm is applicable to any other smooth loss as well. To avoid degeneracy, we constrained each row of $U$ to have unit $\ell_2$ norm. Finally, to pick an appropriate dictionary size, we tried $t \in \{5, 10, 20\}$, which corresponds to under-, perfect- and over-estimation, respectively. The regularization constants $\lambda_1, \lambda_2$ in $\Omega_p$ were chosen from $\{0.01, 0.1, 1, 10, 100\}$.

Note that problem (23) is not jointly convex in $W$ and $U$, so we followed the same strategy as [18]; that is, we alternatively optimized $W$ and $U$ keeping the other fixed. For each subproblem, we ran both APG and GCG to compare their performance. For space limitations, we only report the running time for the setting $\lambda_1 = \lambda_2 = 0.1$, $t = 20$ and $p \in \{1, 2\}$. In these experiments we observed that the polar typically only requires 5 to 6 calls to Prox. As can be seen from Figure 3, GCG is significantly faster than APG and APG-Liu in reducing the objective. This is due to the greedy nature of GCG, which yields very sparse iterates, and when interleaved with local search achieves fast convergence.

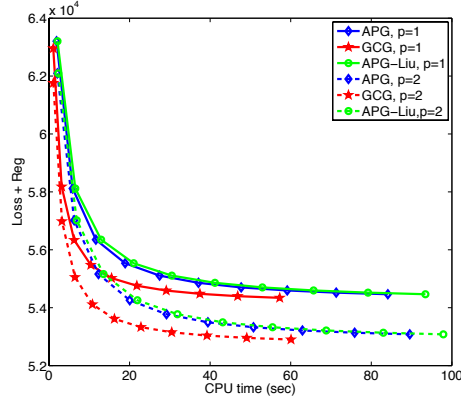

Figure 3: Latent fused lasso.

## 6 Conclusion

We have identified and investigated a new class of structured sparse regularizers whose polar can be reformulated as a linear program with totally unimodular constraints. By leveraging smoothing techniques, we are able to compute the corresponding polars with significantly better efficiency than previous approaches. When plugged into the GCG algorithm, one can observe significant reductions in run time for both group lasso and path coding regularization. We have further developed a generic scheme for converting an efficient proximal solver to an efficient method for computing the polar operator. This reduction allowed us to develop a fast new method for latent fused lasso. For future work, we plan to study more general subset cost functions and investigate new structured regularizers amenable to our approach. It will also be interesting to extend GCG to handle nonsmooth losses.

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
