[Supplementary Material · po--appx.pdf]

# Appendix of *Polar Operators for Structured Sparse Estimation*

## A   Vertices of $Q$ must be scalar multiples of those of $P$

First note that if $\mathbf{0} \notin P$, we have nothing to prove since $P = Q$. Thus we assume $\mathbf{0} \in P$ below.

Consider an arbitrary vertex $\mathbf{q} \in Q$. Clearly $\mathbf{q} \neq \mathbf{0}$ and $\mathbf{q} \in P$, hence $\mathbf{q} = \sum_{i=1}^{n} \alpha_i \cdot \mathbf{p}^{(i)}$, where $n \geq 1$, $\alpha_i > 0$, $\langle \mathbf{1}, \boldsymbol{\alpha} \rangle \leq 1$, and $\mathbf{p}^{(i)}$ are nonzero vertices of $P$. Clearly $\mathbf{p}^{(i)} \in Q$ as $\mathbf{p}^{(i)} \in P$ and $l_i := \langle \mathbf{1}, \mathbf{p}^{(i)} \rangle \geq 1$. It suffices to show $n = 1$. To prove by contradiction, suppose $n \geq 2$.

**(i)** If $\langle \mathbf{1}, \boldsymbol{\alpha} \rangle = 1$, then $\mathbf{q}$ is a convex combination of at least two points in $Q$, hence it cannot be a vertex.

**(ii)** If $\langle \mathbf{1}, \mathbf{q} \rangle = \sum_i \alpha_i l_i = 1$, then $\mathbf{q} = \sum_{i=1}^{n} (\alpha_i l_i) \frac{\mathbf{p}^{(i)}}{l_i}$. But $\frac{\mathbf{p}^{(i)}}{l_i} \in Q$ as $\frac{\mathbf{p}^{(i)}}{l_i} = \frac{1}{l_i} \mathbf{p}^{(i)} + (1 - \frac{1}{l_i}) \mathbf{0} \in P$ and $\langle \mathbf{1}, \mathbf{p}^{(i)} \rangle = l_i \geq 1$. Again contradiction.

**(iii)** If $\langle \mathbf{1}, \mathbf{q} \rangle > 1$ and $\langle \mathbf{1}, \boldsymbol{\alpha} \rangle < 1$, then $\beta := \frac{1}{\langle \mathbf{1}, \mathbf{q} \rangle} < 1 < \frac{1}{\langle \mathbf{1}, \boldsymbol{\alpha} \rangle} =: \gamma$. Clearly $\beta \mathbf{q} \in Q$ because $\beta \mathbf{q} = \beta \mathbf{q} + (1 - \beta) \mathbf{0} \in P$ and $\langle \mathbf{1}, \beta \mathbf{q} \rangle = 1$. Also $\gamma \mathbf{q} \in Q$ as $\gamma \mathbf{q} = \frac{\sum_{i=1}^{n} \alpha_i \mathbf{p}^{(i)}}{\sum_{i=1}^{n} \alpha_i} \in P$ and $\langle \mathbf{1}, \gamma \mathbf{q} \rangle = \frac{\sum_{i=1}^{n} \alpha_i \langle \mathbf{1}, \mathbf{p}^{(i)} \rangle}{\langle \mathbf{1}, \boldsymbol{\alpha} \rangle} \geq \frac{\sum_{i=1}^{n} \alpha_i}{\langle \mathbf{1}, \boldsymbol{\alpha} \rangle} = 1$. So $\mathbf{q}$ lies between two points in $Q$: $\beta \mathbf{q}$ and $\gamma \mathbf{q}$. Contradiction.

Therefore $n = 1$, which completes the proof.

To summarize, we have proved that if $\mathbf{q}$, a vertex of $Q$, is not a vertex of $P$, then it must sum to 1 and be a scalar multiple of some vertex of $P$.

## B   Polar Oracle via Secant Method and Submodular Minimization

By (5), the key optimization problem in computing the polar operator is

$$\lambda^* = \max_{\emptyset \neq A \subseteq [n]} \frac{\langle \tilde{\mathbf{g}}, \mathbf{1}_A \rangle}{F(A)}, \quad \text{where} \quad \tilde{g}_i = |g_i|^p. \tag{26}$$

Let $A^* \in 2^{[n]} \setminus \emptyset$ be a maximizer. The following solution is a slight simplification of [21, §8.4]. Let

$$h(\lambda) := \max_{A \subseteq [n]} \langle \tilde{\mathbf{g}}, \mathbf{1}_A \rangle - \lambda F(A). \tag{27}$$

Note $A = \emptyset$ is allowed here. Clearly $h(\lambda)$ is convex and non-increasing. $h(\lambda) \geq \langle \tilde{\mathbf{g}}, \mathbf{1}_\emptyset \rangle - \lambda F(\emptyset) = 0$. By the definition of $\lambda^*$, for all $A \in 2^{[n]} \setminus \emptyset$ we have $\lambda^* \geq \frac{\langle \tilde{\mathbf{g}}, \mathbf{1}_A \rangle}{F(A)}$, *i.e.* $\langle \tilde{\mathbf{g}}, \mathbf{1}_A \rangle - \lambda^* F(A) \leq 0$. So

$$h(\lambda^*) = \max \left\{ \langle \tilde{\mathbf{g}}, \mathbf{1}_\emptyset \rangle - \lambda^* F(\emptyset), \max_{\emptyset \neq A \subseteq [n]} \langle \tilde{\mathbf{g}}, \mathbf{1}_A \rangle - \lambda^* F(A) \right\} = 0. \tag{28}$$

As a result, $h(\lambda) = 0$ for all $\lambda > \lambda^*$. For any $\lambda < \lambda^*$, $\frac{\langle \tilde{\mathbf{g}}, \mathbf{1}_{A^*} \rangle}{F(A^*)} = \lambda^* > \lambda$, and therefore

$$h(\lambda) \geq \langle \tilde{\mathbf{g}}, \mathbf{1}_{A^*} \rangle - \lambda F(A^*) > 0. \tag{29}$$

In summary,

$$\lambda^* = \sup \{\lambda : h(\lambda) > 0\} = \min \{\lambda : h(\lambda) = 0\}, \tag{30}$$

*i.e.* $\lambda^*$ is the smallest root of $h$, which can be easily found by a secant method thanks to the convexity of $h$. The details are given in Algorithm 2.

Note if $h(\lambda_t) > 0$ upon termination, then the $A_t$ returned must be non-empty. But if $h(\lambda_t) = 0$, then $A_t = \emptyset$ is possible, depending on the solver for the maximization problem in (27). Fortunately, since $\lambda_t = \lambda^*$, it can be easily verified that $\langle \tilde{\mathbf{g}}, \mathbf{1}_{A_{t-1}} \rangle - \lambda^* F(A_{t-1}) = 0$. So we can simply return $A_{t-1}$ without having to customize the solver.

In terms of computational cost, the bottleneck is clearly Step 2 which solves (27) given $\lambda = \lambda_t$. This is deemed as tractable if $F$ is submodular.

**Algorithm 2** Polar oracle via secant method

---

1: Pick arbitrary $A_0 \in [n] \backslash \emptyset$, and set $\lambda_1 = \frac{\langle \tilde{\mathbf{g}}, \mathbf{1}_{A_0} \rangle}{F(A_0)}$. Clearly $\lambda_1 \leq \lambda^*$ and so $h(\lambda_1) \geq 0$.

2: **for** $t = 1, 2, \ldots$ **do**

3:     Compute $h(\lambda_t)$ by finding an optimal $A$ in the definition of $h(\lambda_t)$ in (27). Call this $A$ as $A_t$.

4:     **if** $h(\lambda_t) \in (0, \epsilon)$ **then**

5:         **return** $A_t$. $\lambda_\epsilon^* = \frac{\langle \tilde{\mathbf{g}}, \mathbf{1}_{A_t} \rangle}{F(A_t)}$ can be at most $\epsilon$ smaller than the true $\lambda^*$.

6:     **end if**

7:     **if** $h(\lambda_t) = 0$ **then**

8:         **return** $A_{t-1}$. It follows that $\lambda^* = \frac{\langle \tilde{\mathbf{g}}, \mathbf{1}_{A_{t-1}} \rangle}{F(A_{t-1})}$.

9:     **end if**

10:     Linearize $h(\lambda)$ at $\lambda_t$ as $\tilde{h}_t(\lambda) = h(\lambda_t) - (\lambda - \lambda_t)F(A_t)$.

11:     Set $\lambda_{t+1}$ as the root of $\tilde{h}_t$: $\lambda_{t+1} = \lambda_t + \frac{h(\lambda_t)}{F(A_t)} = \lambda_t + \frac{\langle \tilde{\mathbf{g}}, \mathbf{1}_{A_t} \rangle - \lambda_t F(A_t)}{F(A_t)} = \frac{\langle \tilde{\mathbf{g}}, \mathbf{1}_{A_t} \rangle}{F(A_t)}$. Since $h$ is convex and hence $\tilde{h}_t$ must be upper bounded by $h$, it follows $\lambda_{t+1} \leq \lambda^*$. Thus $h(\lambda_{t+1}) \geq 0$.

12: **end for**

---

## B.1  Network Min-cut Algorithm for Submodular Minimization with Overlapping Group

Next we show the network max-flow/min-cut algorithm for solving (27) in overlapping group lasso. Using the notation and setup in Section 3.1, the problem (27) can be written as

$$\min_{\mathbf{w} \in \{0,1\}^{n+l}} \lambda \sum_{G \in \mathcal{G}} c_G w_G - \sum_{i \in [n]} \tilde{g}_i w_i, \quad s.t. \quad w_G \geq w_i, \ \forall i \in G \in \mathcal{G}. \tag{31}$$

This is obviously equivalent to

$$\min_{\mathbf{w} \in \{0,1\}^{n+l}} \sum_{G \in \mathcal{G}} (\lambda c_G) w_G + \sum_{i \in [n]} \tilde{g}_i (1 - w_i), \quad s.t. \quad w_G \geq w_i, \ \forall i \in G \in \mathcal{G}. \tag{32}$$

Now we show this is exactly a min-cut problem on a directed graph. Let us construct a directed graph with source node $s$, sink node $t$, and all nodes $w_G$ and $w_i$. There is a directed edge from $s$ to each node $w_G$ ($G \in \mathcal{G}$), and the weight is $\eta_G := \lambda c_G$. In addition, there is a directed edge from each node $w_i$ ($i \in [n]$) to the sink $t$, with weight $\eta_i := \tilde{g}_i$. Finally, for each $i \in G \in \mathcal{G}$, there is a edge from node $w_G$ to $w_i$, and the weight is $\eta_{G,i} := \infty$.

The min-cut problem essentially divides all nodes in a graph into two groups $S$ and $T$ with $s \in S$ and $t \in T$, and minimizes the sum of the weight of all edges from $u$ to $v$ where $u \in S$ and $v \in T$. Note edges with $u \in T$ and $v \in S$ are not counted into the cut-edge by the definition of min-cut. Let us fix $p_s = 0$, $p_t = 1$, and use $p_i, p_G = 0$ (or 1) if the node belongs to $S$ (or $T$). Then the min-cut objective for this directed graph can be written as

$$\min_{p_i, p_G \in \{0,1\}} \sum_{i \in G \in \mathcal{G}: p_G = 0, p_i = 1} \eta_{G,i} + \sum_{i \in [n]: p_i = 0} \eta_i + \sum_{G \in \mathcal{G}: p_G = 1} \eta_G. \tag{33}$$

Since $\eta_{G,i} = \infty$, we have to exclude the solutions where $p_G = 0$ and $p_i = 1$. This can be compactly enforced by adding constraints $p_G \geq p_i$. Moreover, it is obvious from $p_i, p_G \in \{0,1\}$ that

$$\sum_{i \in [n]: p_i = 0} \eta_i = \sum_{i \in [n]} \eta_i (1 - p_i), \quad \text{and} \quad \sum_{G \in \mathcal{G}: p_G = 1} \eta_G = \sum_{G \in \mathcal{G}} \eta_G p_G. \tag{34}$$

Substituting them back into (33) and noting the definition of $\eta_i$ and $\eta_G$, it is straightforward to observe the equivalence between (32) and (33), with $p_G$ and $p_i$ corresponding to $w_G$ and $w_i$ respectively.

Finally, by using the well-known equivalence between max-flow and min-cut (problem (33)), it is trivial to write out the max-flow formulation for the graph defined above, which exactly recovers the solution proposed by [11, Algorithm 2]. In comparison, our min-cut formulation is clearly more straightforward because it completely eliminates the dualization step and directly provides the solution to (27).

## C  Proof of Proposition 1

The proof is based on the well-known duality between strong convexity and smoothness (Lipschitz continuous gradient) [17]. Note that we assume that $r$, the upper bound on the number of groups each variable can belong to, is greater than 1 since otherwise the problem is trivial.

*Proof:*  Note that there are $n$ variables which we index by $i$ and there are $\ell$ groups (subsets of variables) which we index by $G$. The input vector $\tilde{\mathbf{w}} \in \mathbb{R}^n \times \mathbb{R}^\ell$.

Let $l_i$ be the number of groups that contain variable $i$, and $\mathcal{S}_i := \{\mathbf{s} \in \mathbb{R}_+^{l_i} : \langle \mathbf{1}, \mathbf{s} \rangle = 1\}$ be the $(l_i - 1)$-dimensional simplex. Using the well-known variational representation of max function, we rewrite the (negated) objective $h$ in (15) as

$$h(\tilde{\mathbf{w}}) = \sum_{i \in [n]} \tilde{g}_i \max_{\boldsymbol{\alpha}^{(i)} \in \mathcal{S}_i} \left\{ -\sum_{G:i \in G} \alpha_G^{(i)} \tilde{w}_G \right\} = \max_{\boldsymbol{\alpha}^{(i)} \in \mathcal{S}_i} \sum_{i \in [n]} \sum_{G:i \in G} -\tilde{g}_i \tilde{w}_G \alpha_G^{(i)}, \qquad (35)$$

which is to be minimized. Here the second equality follows from the separability of the variables $\boldsymbol{\alpha}^{(i)}$. Fix $\epsilon > 0$ and denote $c := \frac{\epsilon}{n \log r}$. Consider

$$h_\epsilon(\tilde{\mathbf{w}}) = \max_{\boldsymbol{\alpha}^{(i)} \in \mathcal{S}_i} \sum_{i \in [n]} \sum_{G:i \in G} \left( -\tilde{g}_i \tilde{w}_G \alpha_G^{(i)} - c \cdot \alpha_G^{(i)} \log \alpha_G^{(i)} \right),$$

*i.e.*, we add to $h$ the scaled entropy function $-c \sum_{i \in [n], G:i \in G} \alpha_G^{(i)} \log \alpha_G^{(i)}$ whose negation is known to be strongly convex on the simplex (*w.r.t.* the $\ell_1$-norm) [17]. Since the entropy is nonnegative, we have for any $\tilde{\mathbf{w}}$, $h(\tilde{\mathbf{w}}) \leq h_\epsilon(\tilde{\mathbf{w}})$ and moreover

$$h_\epsilon(\tilde{\mathbf{w}}) - h(\tilde{\mathbf{w}}) \leq c \max_{\boldsymbol{\alpha}^{(i)} \in \mathcal{S}_i} \sum_{i \in [n]} \sum_{G:i \in G} -\alpha_G^{(i)} \log \alpha_G^{(i)} \leq c \cdot n \log r = \epsilon,$$

where the last inequality is due to the well-known upper bound of the entropy over the probability simplex, *i.e.* entropy attains its maximum when all odds are equally likely. Therefore $h(\tilde{\mathbf{w}}) - h_\epsilon(\tilde{\mathbf{w}}) \in (-\epsilon, 0]$, and we have proved part (ii) of Proposition 1.

By straightforward calculation

$$h_\epsilon(\tilde{\mathbf{w}}) = \sum_{i \in [n]} \max_{\boldsymbol{\alpha}^{(i)} \in \mathcal{S}_i} \sum_{G:i \in G} \left( -\tilde{g}_i \tilde{w}_G \alpha_G^{(i)} - c \cdot \alpha_G^{(i)} \log \alpha_G^{(i)} \right)$$

$$= c \sum_{i \in [n]} \log \sum_{G:i \in G} \exp \left( -\frac{\tilde{g}_i \tilde{w}_G}{c} \right), \qquad (36)$$

$$\frac{\partial}{\partial \tilde{w}_G} h_\epsilon(\tilde{\mathbf{w}}) = -\sum_{i:i \in G} \tilde{g}_i p_i(G), \quad \text{where} \quad p_i(G) := \frac{\exp\left( -\frac{\tilde{g}_i \tilde{w}_G}{c} \right)}{\sum_{\tilde{G}:i \in \tilde{G}} \exp\left( -\frac{\tilde{g}_i \tilde{w}_{\tilde{G}}}{c} \right)}. \qquad (37)$$

Hence $h_\epsilon(\tilde{\mathbf{w}})$ can be computed in $O(nr)$ time (since the second summation in (36) contains at most $r$ terms). Similarly all $\{p_i(G) : i \in [n], i \in G\}$ can be computed in $O(nr)$ time. Therefore part (iii) of Proposition 1 is established.

Finally, to bound the Lipschitz constant of the gradient of $h_\epsilon$, we observe that $h_\epsilon(\tilde{\mathbf{w}}) = \eta^*(A\tilde{\mathbf{w}})$, where $\eta^*$ is the Fenchel conjugate of the scaled negative entropy

$$\eta(\boldsymbol{\alpha}) = c \sum_{i \in [n]} \sum_{G:i \in G} \alpha_G^{(i)} \log \alpha_G^{(i)},$$

and $A$ is defined as the matrix satisfying

$$\langle \boldsymbol{\alpha}, A\tilde{\mathbf{w}} \rangle = \sum_{i \in [n]} \sum_{G:i \in G} -\alpha_G^{(i)} \tilde{g}_i \tilde{w}_G.$$

It is known that the scaled negative entropy $\eta$ is strongly convex with modulus $c$ (*w.r.t.* the $\ell_1$-norm). Furthermore, employing $\ell_1$ norm on $\boldsymbol{\alpha}$ and $\ell_2$ norm on $\tilde{\mathbf{w}}$, the operator norm of the matrix $A$ can be

bounded as

$$\|A\|_{2,1} := \max_{\boldsymbol{\alpha}:\|\boldsymbol{\alpha}\|_1=1} \max_{\tilde{\mathbf{w}}:\|\tilde{\mathbf{w}}\|_2=1} \langle \boldsymbol{\alpha}, A\tilde{\mathbf{w}} \rangle = \max_{\tilde{\mathbf{w}}:\|\tilde{\mathbf{w}}\|_2=1} \max_{\boldsymbol{\alpha}:\|\boldsymbol{\alpha}\|_1=1} \sum_{i\in[n]} \sum_{G:i\in G} -\alpha_G^{(i)} \tilde{g}_i \tilde{w}_G \quad (38)$$

$$\leq \left( \max_{i\in[n]} \tilde{g}_i \right) \cdot \max_{\tilde{\mathbf{w}}\geq 0:\|\tilde{\mathbf{w}}\|_2=1} \max_{\boldsymbol{\alpha}\geq 0:\|\boldsymbol{\alpha}\|_1=1} \sum_{i\in[n]} \sum_{G:i\in G} \alpha_G^{(i)} \tilde{w}_G \quad (39)$$

$$\leq \left( \max_{i\in[n]} \tilde{g}_i \right) \cdot \max_{\boldsymbol{\alpha}\geq 0:\|\boldsymbol{\alpha}\|_1=1} \sum_{i\in[n]} \sum_{G:i\in G} \alpha_G^{(i)} = \max_{i\in[n]} \tilde{g}_i = \|\tilde{\mathbf{g}}\|_\infty . \quad (40)$$

The equality is obviously attainable. Therefore by Theorem 1 of [17], $h_\epsilon(\tilde{\mathbf{w}}) = \eta^*(A\tilde{\mathbf{w}})$ has Lipschitz continuous gradient *w.r.t.* $\ell_2$ norm, and the Lipschitz constant is

$$\frac{1}{c} \|A\|_{2,1}^2 = \frac{1}{\epsilon} \|\tilde{\mathbf{g}}\|_\infty^2 \, n \log r.$$

This completes our proof of part (i) of Proposition 1. ∎

# D  DAG Groups

We discuss here another interesting special case of the group sparse model formulated in Section 3.1.

Suppose the variables $\{1, 2, \ldots, n\}$ form the nodes of a directed acyclic graph (DAG), and each node $i$ corresponds to a group consisting of all nodes $j$ that are reachable from $i$ by transversing the DAG. For simplicity we assign unit cost to each group. Since a node in this model may belong to $n$ groups, *i.e.* $r = \Theta(n)$ (recall that $r$ is the upper bound on the number of groups that any variable may belong to), hence a naive application of Proposition 1 results in the overall complexity for computing the polar as $O(\frac{1}{\epsilon}\sqrt{n^5 \log n})$. Fortunately this can be reduced to $O(\frac{1}{\epsilon}m\sqrt{n})$, where $m$ is the number of edges (in the worst case on the order of $n^2$).

We recall from the main paper the polar of the general group sparse regularizer

$$\min_{\tilde{\mathbf{w}}\geq 0} \sum_{i\in[n]} \tilde{g}_i \cdot \max_{G:i\in G\in\mathcal{G}} (-\tilde{w}_G), \ \text{ s.t. } \sum_{G\in\mathcal{G}} b_G \cdot \tilde{w}_G = 1.$$

In the DAG case, each variable $i$ corresponds to a group that consists of all descendants of $i$. Let us denote the group as $G_i$. For simplicity, assume the costs $b_G = 1$ for all groups $G$. By symmetry, if there is an edge from $i$ to $j$ then at optimum $\tilde{w}_{G_i} \geq \tilde{w}_{G_j}$, because otherwise we can swap their values without increasing the objective or violating the constraint. To lighten notation, we just write $\tilde{w}_{G_i}$ as $\tilde{w}_i$. Thus we simplify the above problem into

$$\min_{\tilde{\mathbf{w}}\geq 0} - \sum_{i\in[n]} \tilde{g}_i \tilde{w}_i, \ \text{ s.t. } \sum_{i\in[n]} \tilde{w}_i = 1, \text{ and } \tilde{w}_i \geq \tilde{w}_j \ \forall \, (i,j) \in E. \quad (41)$$

Here we use the pair $(i, j)$ to denote an edge from $i$ to $j$, and $E$ is the set of all edges. Next introduce the dual variables $\alpha_{ij} \geq 0$ for the constraint $\tilde{w}_i \geq \tilde{w}_j$ and $\xi$ for the constraint $\sum_{i\in[n]} \tilde{w}_i = 1$. Consider the Lagrangian dual

$$\min_{\xi,\boldsymbol{\alpha}\geq\mathbf{0}} \xi + \sum_{i\in[n]} \max_{\tilde{w}_i\geq 0} \tilde{w}_i \left( \tilde{g}_i - \xi + \sum_{j:(i,j)\in E} \alpha_{ij} - \sum_{k:(k,i)\in E} \alpha_{ki} \right),$$

which, after taking into account $\tilde{w}_i \leq 1$, simplifies to

$$\min_{\xi,\boldsymbol{\alpha}\geq\mathbf{0}} \xi + \sum_{i\in[n]} \left( \tilde{g}_i - \xi + \sum_{j:(i,j)\in E} \alpha_{ij} - \sum_{k:(k,i)\in E} \alpha_{ki} \right)_+ , \quad (42)$$

where $(x)_+ := \max\{x, 0\}$. As in Appendix C we can easily smooth the function $(\cdot)_+$ and therefore solve (42) using APG. To summarize, a $2\epsilon$ accurate solution can be found in $O(\frac{1}{\epsilon}\sqrt{n})$ iterations

with $O(m)$ cost per iteration. Overall this is faster than the complexity $O(mn^2 \log \frac{1}{\epsilon})$ of [6] (which involves a binary search). See Appendix E for details.

Moreover, if the DAG is a rooted tree, *i.e.*, each node can only be pointed to by at most one edge, we can further reduce the overall cost to $O(n \log \frac{1}{\epsilon})$. Indeed, let the root be node 1, and denote as $\mathrm{pa}(i)$ and $\mathrm{ch}(i)$ the parent and children nodes of $i$, respectively. Note that by the definition of rooted tree, $|\mathrm{pa}(i)| = 1$ for any node $i$ that is not the root. Again, for any non-root node $i > 1$, we introduce a dual variable $\alpha_i$ for the constraint $x_{\mathrm{pa}(i)} \geq x_i$. For convenience let $\alpha_1 = 0$. Then the Lagrangian dual of (41) in the rooted tree case is

$$\min_{\xi, \boldsymbol{\alpha} \geq 0} \xi + \sum_{i \in [n]} \left( \tilde{g}_i - \xi + \sum_{j \in \mathrm{ch}(i)} \alpha_j - \alpha_i \right)_+ . \tag{43}$$

At the optimum, there cannot be two summands that are positive, because then the subgradient of $\xi$ would be negative. If only one summand is positive, we can increase $\xi$ to make it 0 without changing the objective value. Thus we can assume all summands are 0, and solve

$$\min_{\xi, \boldsymbol{\alpha} \geq 0} \xi, \ \text{s.t.} \ \forall i, \ \alpha_i \geq \tilde{g}_i - \xi + \sum_{j \in \mathrm{ch}(i)} \alpha_j. \tag{44}$$

In effect, we search for the smallest $\xi$ that makes the feasible region nonempty. For any $\xi > 0$, its feasibility can be checked by propagating towards the root via

$$\alpha_i = \max \left\{ 0, \ \tilde{g}_i - \xi + \sum_{j \in \mathrm{ch}(i)} \alpha_j \right\}. \tag{45}$$

Note that for all leaf nodes, that is $\{j : \mathrm{ch}(j) = \emptyset\}$, their dual variables $\alpha_j = 0$. At the root if $\alpha_1 = 0 \geq \tilde{g}_1 - \xi + \sum_{j \in \mathrm{ch}(1)} \alpha_j$ is met, then we claim that $\xi$ is feasible. Clearly $\xi \in [\tilde{g}_1, \max_i \tilde{g}_i]$, hence using binary search an $\epsilon$ accurate solution can be found in $O(n \log \frac{1}{\epsilon})$. Finally, given $\xi$, the optimal primal variable $\tilde{\mathbf{w}}$ can be easily recovered using KKT conditions. Overall our approach is faster than the $O(nd)$ complexity in [5], where d is the depth of the tree and in the worst case can be $\Theta(n)$.

# E  Comparisons for Group Sparse Models

In this section we compare the complexity of our approach (under the group sparse model developed in Section 3.1) with two related methods in literature, namely, [11] and [6].

Consider first [11]. The Algorithm 2 there proceeds in loops, with each iteration involving a max-flow problem on the canonical graph. The loop can take at most $n$ iterations, while each max-flow problem can be solved with $O(|V| |E|)$ cost where $|V|$ and $|E|$ are the number of nodes and edges in the canonical graph, respectively. By construction, $|V| = n + l$, and $|E| \leq nr$ since each pair of $(G, i)$ with the node $i$ belong to the group $G$ contributes an edge. Therefore the total cost is upper bounded by $O(n^2(n + l)r)$. Note that in the worst case $\ell = \Theta(nr)$. In contrast, the approach we developed in Section 3.1 for bounded degree groups costs $O(\frac{nr}{\epsilon}\sqrt{n \log r})$, significantly cheaper in the regime where $n$ is big and $\epsilon$ is moderate.

For the DAG groups considered in Appendix D, again Algorithm 2 in [11] can take $\Theta(n)$ iterations, while $|V| = 2n$ and $|E| \leq mn$ (since in the worst case each node can belong to $\Theta(n)$ groups). Thus overall [11, Algorithm 2] costs $O(n^3 m)$ for DAG groups, worse than the complexity $O(\frac{1}{\epsilon} m \sqrt{n})$ we obtained in Appendix D.

Next consider [6] which developed a line search scheme to compute the polar. The major computational step there is to solve

$$\tilde{\mathbf{w}}_\sigma = \arg\max_{\tilde{\mathbf{w}} \in Q} \langle \tilde{\mathbf{g}}, \tilde{\mathbf{w}} \rangle - \sigma \langle \mathbf{b}, \tilde{\mathbf{w}} \rangle, \tag{46}$$

recursively, each time with a updated $\sigma > 0$. In the case of bounded degree groups, this is again a max-flow problem which costs $O(n(n+l)r)$, and therefore the overall cost is $O(n(n+l)r \log \frac{1}{\epsilon})$. In the case of DAG groups (Appendix D), the max-flow problem costs $O(n^2 m)$, and hence the overall cost is $O(n^2 m \log \frac{1}{\epsilon})$. In both cases, [6] improves over [11] but is still worse than our approach.

# F Path Coding: Efficient Linear Programming

We show in this section how to efficiently solve the LP for the path coding regularizer discussed in Section 3.2. First recall that we have arrived at the following LP in Section 3.2:

$$\max_{\tilde{\mathbf{w}}} \quad \sum_i \tilde{g}_i \left( \sum_{j:(i,j)\in E} \tilde{w}_{ij} + \sum_{k:(k,i)\in E} \tilde{w}_{ki} \right), \tag{47}$$

$$\text{s.t.} \quad \tilde{\mathbf{w}} \geq \mathbf{0}, \quad \sum_{(i,j)\in E} b_{ij}\tilde{w}_{ij} = 1, \quad \sum_{j:(i,j)\in E} \tilde{w}_{ij} = \sum_{k:(k,i)\in E} \tilde{w}_{ki}, \; \forall i. \tag{48}$$

This LP appears to be more complicated than the one in Section 3.1, due to the two extra constraints in the end. We start with removing these constraints by introducing dual variables.

Denote $z_i = \sum_{j:(i,j)\in E} \tilde{w}_{ij}$. Since $\tilde{w}_{ij} \geq 0$, we can parameterize $\tilde{w}_{ij}$ as $\tilde{w}_{ij} = z_i \tau_j^{(i)}$, where $z_i \geq 0$ and $\boldsymbol{\tau}^{(i)}$ belongs to the simplex $\mathcal{S}_i := \{\boldsymbol{\tau}^{(i)} \geq \mathbf{0} : \langle \mathbf{1}, \boldsymbol{\tau}^{(i)} \rangle = 1\}$. Introduce Lagrange multipliers $\vartheta = (\lambda, \alpha_i)$ for the three constraints in (47), respectively. For convenience also let $\alpha_s = \alpha_t = \tilde{g}_s = \tilde{g}_t = 0$. Denote

$$d_{ij}(\vartheta) = \tilde{g}_i + \tilde{g}_j - \alpha_i + \alpha_j - \lambda b_{ij}.$$

After some tedious algebra we obtain the Lagrangian

$$\min_{\boldsymbol{\alpha},\lambda} \left\{ \lambda + \sum_{(i,j)\in E} \max_{\tilde{w}_{ij}\geq 0} \tilde{w}_{ij} d_{ij}(\vartheta) \right\} = \min_{\boldsymbol{\alpha},\lambda} \left\{ \lambda + \sum_{i\in[n]\cup\{s\}} \max_{z_i\geq 0} z_i \max_{\boldsymbol{\tau}^{(i)}\in\mathcal{S}_i} \sum_{j:(i,j)\in E} \tau_j^{(i)} d_{ij}(\vartheta) \right\} \tag{49}$$

$$= \min_{\boldsymbol{\alpha},\lambda} \left\{ \lambda + \sum_{i\in[n]\cup\{s\}} \max_{z_i\geq 0} z_i \left( \max_{j:(i,j)\in E} d_{ij}(\vartheta) \right) \right\}. \tag{50}$$

Our key observation is that $z_i$ can be upper bounded. Note the constraints $\sum_{(i,j)\in E} b_{ij}\tilde{w}_{ij} = 1$ and $\tilde{\mathbf{w}} \geq 0$ in (48). Let $C$ be the lowest cost of all $(s,t)$-paths, and naturally $C > 0$ by assumption. Then trivially any path will satisfy $z_i \leq \rho := \frac{1}{C}$. A more conservative upper bound on $z_i$ is

$$z_i \leq \rho := \left( \min_{(i,j)\in E} b_{ij} \right)^{-1}, \tag{51}$$

assuming all $b_{ij} > 0$. Taking into account these upper bounds, we arrive at our final objective

$$\min_{\lambda} \{\lambda + \rho f(\lambda)\}, \quad \text{where} \quad f(\lambda) := \min_{\boldsymbol{\alpha}} \sum_{i\in[n]\cup\{s\}} \left( \max_{j:(i,j)\in E} d_{ij}(\vartheta) \right)_+. \tag{52}$$

As before $(x)_+ = \max\{x, 0\}$. Note given $\lambda$, the inner optimization over $\boldsymbol{\alpha}$ has a closed form thanks to the absence of cycles. Specifically, let $\alpha_t(\lambda) = 0$ and define for any $i \in [n] \cup \{s\}$

$$\alpha_i(\lambda) = \max_{j:(i,j)\in E} \{\alpha_j(\lambda) + \tilde{g}_i + \tilde{g}_j - b_{ij}\lambda\}. \tag{53}$$

Since the graph is a DAG, we can always find a topological ordering of the indices $i$, such that before computing $\alpha_i(\lambda)$ for node $i$, all its descendants $\alpha_j(\lambda)$ have been computed. It is not hard to see

$$f(\lambda) = \max\{\alpha_s(\lambda), 0\}, \tag{54}$$

and the optimal $\boldsymbol{\alpha}$ in the definition of $f$ in (52) is attained at $\{\alpha_i = \alpha_i(\lambda) : i \in [n]\}$, because, as can be easily verified, $\mathbf{0}$ is a subgradient. This relationship allows us to compute a subgradient of $f$ at $\lambda$ via recursion

$$\partial\alpha_i(\lambda) = \left\{ \sum_{j\in J} \gamma_j (v_j - b_{ij}) : J = (\text{set of}) \arg\max \text{ in } (53), v_j \in \partial\alpha_j(\lambda), \gamma_j \geq 0, \langle \mathbf{1}, \boldsymbol{\gamma} \rangle = 1 \right\}. \tag{55}$$

Obviously, the recursion in both (53) and (55) can be accomplished in $O(m)$ time. Indeed a trivial subgradient of $\alpha_s(\lambda)$ is the negative cost of the path that is induced by the $\arg\max$ in (53) (breaking

tie arbitrarily). Finally we solve (52) over $\lambda$ by cutting plane method, which can find an $\epsilon$ accurate solution in $O(\frac{n}{\epsilon^2})$ iterations, *i.e.* with $O(\frac{mn}{\epsilon^2})$ total computation.

Further reducing the computational cost to $O(\frac{m\sqrt{n}}{\epsilon})$ is possible by smoothing the $\max$ function in

$$\min_{\boldsymbol{\alpha},\lambda} \left\{ \lambda + \rho \sum_{i\in[n]\cup\{s\}} \left( \max_{j:(i,j)\in E} d_{ij}(\vartheta) \right)_+ \right\}. \tag{56}$$

This cost is potentially better than the $O(mn)$ worst case complexity in [12, Algorithm 1]. Algorithmically, this can be done in exactly the same way as in Appendix C. After that we run APG on the smoothed problem. To summarize, following exactly the same argument as in the proof of Proposition 1 we have

**Proposition 4** *Denote the objective in (56) as $h(\vartheta)$, For any $\epsilon > 0$, there exists a convex function $h_\epsilon$ such that (i) $\forall \vartheta$, $h(\vartheta) - h_\epsilon(\vartheta) \in (-\epsilon, 0]$, (ii) $h_\epsilon$ has $L = O(\frac{n}{\epsilon})$ Lipschitz continuous gradient, and (iii) the gradient of $h_\epsilon$ can be computed in $O(m)$ time.*

# G   Recovery of Integral Solutions to Polar Oracle

Recall our ultimate goal in polar oracle is to find integral solutions to (8) which we copy here for convenience

$$\lambda^* := \max_{\mathbf{0}\neq\mathbf{w}\in P} \frac{\langle \tilde{\mathbf{g}}, \mathbf{w} \rangle}{\langle \mathbf{b}, \mathbf{w} \rangle}. \tag{57}$$

As we showed in Section 3, the optimal objective value is exactly equal to that of (10), which we also copy here

$$\max_{\tilde{\mathbf{w}},\sigma>0} \ \langle \tilde{\mathbf{g}}, \tilde{\mathbf{w}} \rangle, \ \text{subject to} \ \tilde{\mathbf{w}} \in \sigma Q, \ \langle \mathbf{b}, \tilde{\mathbf{w}} \rangle = 1. \tag{58}$$

We have shown how to smooth this objective and find an $\epsilon$ accurate solution for it. That means we have obtained a $\lambda_\epsilon$ (smooth objective function value) with the guarantee that $\lambda_\epsilon \in [\lambda^* - \epsilon, \lambda^*]$. With this $\lambda_\epsilon$ in hand, we now show how to find an $\epsilon$ accurate solution for (8), *i.e.* a $\mathbf{w}_\epsilon \in P\backslash\{\mathbf{0}\}$ such that

$$\frac{\langle \tilde{\mathbf{g}}, \mathbf{w}_\epsilon \rangle}{\langle \mathbf{b}, \mathbf{w}_\epsilon \rangle} \geq \lambda^* - \epsilon. \tag{59}$$

Indeed, this is simple according to Proposition 5.

**Proposition 5** *Given $\lambda_\epsilon \in [\lambda^* - \epsilon, \lambda^*]$, find*

$$\mathbf{w}_\epsilon := \arg\max_{\mathbf{w}\in P\backslash\{\mathbf{0}\}} \left\{ \langle \tilde{\mathbf{g}}, \mathbf{w} \rangle - \lambda_\epsilon \langle \mathbf{b}, \mathbf{w} \rangle \right\}. \tag{60}$$

*Then $\mathbf{w}_\epsilon$ must satisfy (59).*

*Proof:*   By the definition of $\lambda^*$, $\max_{\mathbf{w}\in P\backslash\{\mathbf{0}\}} \{ \langle \tilde{\mathbf{g}}, \mathbf{w} \rangle - \lambda^* \langle \mathbf{b}, \mathbf{w} \rangle \} = 0$. As $\lambda_\epsilon \leq \lambda^*$, so $\max_{\mathbf{w}\in P\backslash\{\mathbf{0}\}} \{ \langle \tilde{\mathbf{g}}, \mathbf{w} \rangle - \lambda_\epsilon \langle \mathbf{b}, \mathbf{w} \rangle \} \geq 0$. This implies $\frac{\langle \tilde{\mathbf{g}}, \mathbf{w}_\epsilon \rangle}{\langle \mathbf{b}, \mathbf{w}_\epsilon \rangle} \geq \lambda_\epsilon \geq \lambda^* - \epsilon$. ∎

Note (60) is exactly the submodular minimization problem that the secant method is based on (step 3 of Algorithm 2). This step is computationally expensive and has to be solved for multiple values of $\lambda_t$ in that method. In contrast, our our strategy needs to solve this problem only once.

In group sparsity, it leads to a max-flow problem as in Appendix B.1 which is again expensive. Fortunately, by exploiting the structure of the problem it is possible to design a heuristic solution. For convenience let us copy (15) to here, the linear programming for group sparsity.

$$\max_{\tilde{\mathbf{w}}} \ \sum_{i\in[n]} \tilde{g}_i \min_{G:i\in G\in\mathcal{G}} \tilde{w}_G, \ \text{subject to} \ \tilde{\mathbf{w}} \geq 0, \ \sum_{G\in\mathcal{G}} b_G \tilde{w}_G = 1. \tag{61}$$

A solution $\tilde{\mathbf{w}}$ corresponds to an integral solution to the polar oracle if and only if $\tilde{w}_G \in \{0, c\}$ where $c$ ensures $\sum_{G \in \mathcal{G}} b_G \tilde{w}_G = 1$. By solving the smoothed objective, we obtain a solution $\tilde{\mathbf{w}}^*$ which does not necessarily satisfy this condition. However, a smaller value of the component $\tilde{w}_G^*$ does suggest a higher likelihood for $\tilde{w}_G$ to be 0. Therefore, we sorted $\{w_G^*\}$ and set the $w_G$ of the smallest $k$ groups to 0 ($k$ ranging from 0 to $|\mathcal{G}| - 1$), and the $w_G$ for the remaining groups were set to a common value that satisfies the constraint. Given $k$, this leads to an objective value, and the $k$ that maximizes this value can be selected by enumerating $k \in \{0, 1, \ldots, |\mathcal{G}| - 1\}$. By exploiting the structure of the objective, it is easy to design an algorithm which accomplishes the enumeration in $O(nr)$ time.

The optimal objective value over all $k$ also allows us to compute its distance to the optimal objective value of the smoothed objective. If the gap (used as a certificate) is below $\epsilon$, this integral solution is exactly $\epsilon$ sub-optimal. Otherwise we fall back on (60), and this case rarely happens in practice.

In path coding, the path can be simply recovered by following the $\arg\max$ in (53), with $\lambda$ set to an optimal solution to (52).

# H  Polar of $\Omega_s(\mathbf{w}) = \sum_i \|\mathbf{w}\|_{(i)}$

The polar of $\Omega_s(\mathbf{w}) = \sum_i \|\mathbf{w}\|_{(i)}$ follows from the following proposition by taking $\phi(\boldsymbol{\alpha}) = \sum_i \alpha_i$. We note that Proposition 6 itself is a slight generalization of [19, Theorem 15.3].

**Proposition 6** *Let* $\kappa_i : \mathbb{R}^d \to \bar{\mathbb{R}}_+, 1 \leq i \leq n$ *be closed gauges,* $\phi : \bar{\mathbb{R}}_+^n \to \bar{\mathbb{R}}_+$ *be closed, convex, non-constant in each coordinate[1] with* $\phi(\mathbf{0}) = 0$, *and* $\exists \mathbf{x} \in \cap_i \mathrm{ri}\,\mathrm{dom}\,\kappa_i$ *such that* $(\kappa_1(\mathbf{x}), \ldots, \kappa_n(\mathbf{x})) \in \mathrm{ri}\,\mathrm{dom}\,\phi$, *then the Fenchel conjugate of* $h := \phi(\kappa_1, \ldots, \kappa_n)$ *is*

$$h^*(\mathbf{x}) = \min_{\sum_i \mathbf{x}^i = \mathbf{x}} \phi^+(\kappa_1^\circ(\mathbf{x}^1), \ldots, \kappa_n^\circ(\mathbf{x}^n)), \tag{62}$$

*where* $\kappa_i^\circ$ *is the polar of* $\kappa_i$ *and* $\phi^+(\mathbf{y}) := \max_{\mathbf{x} \geq 0} \langle \mathbf{x}, \mathbf{y} \rangle - \phi(\mathbf{x})$ *is the monotone conjugate of* $\phi$. *Moreover, if* $\phi$ *is a gauge so is* $h$ *whose polar*

$$h^\circ(\mathbf{x}) = \min_{\sum_i \mathbf{x}^i = \mathbf{x}} \phi^\circ(\kappa_1^\circ(\mathbf{x}^1), \ldots, \kappa_n^\circ(\mathbf{x}^n)), \tag{63}$$

*where* $\phi^\circ$ *is the polar of* $\phi$.

*Proof:* Let us define the diagonal operator $A : \bar{\mathbb{R}}^d \to (\bar{\mathbb{R}}^d)^n, \mathbf{x} \mapsto (\mathbf{x}, \ldots, \mathbf{x})$. Then $h(\mathbf{x}) = H(A(\mathbf{x}))$, where

$$H(\mathbf{x}^1, \ldots, \mathbf{x}^n) := \phi(\kappa_1(\mathbf{x}^1), \ldots, \kappa_n(\mathbf{x}^n)).$$

The Fenchel conjugate of $G$ is

$$
\begin{aligned}
H^*(\mathbf{y}^1, \ldots, \mathbf{y}^n) &= \sup_{\mathbf{x}^i} \sum_i \langle \mathbf{x}^i, \mathbf{y}^i \rangle - H(\mathbf{x}^1, \ldots, \mathbf{x}^n) \\
&= \sup_{\mathbf{x}^i} \sum_i \langle \mathbf{x}^i, \mathbf{y}^i \rangle - \phi(\kappa_1(\mathbf{x}^1), \ldots, \kappa_n(\mathbf{x}^n)) \\
&= \sup_{\kappa_i(\mathbf{x}^i) \leq \lambda_i} \sum_i \langle \mathbf{x}^i, \mathbf{y}^i \rangle - \phi(\lambda_1, \ldots, \lambda_n) \\
&= \sup_{\lambda_i \geq 0} \sum_i \langle \kappa_i^\circ(\mathbf{y}^i), \lambda_i \rangle - \phi(\lambda_1, \ldots, \lambda_n) \\
&= \phi^+(\kappa_1^\circ(\mathbf{y}^1), \ldots, \kappa_n^\circ(\mathbf{y}^n)),
\end{aligned}
$$

where the third equality is due to the monotonicity of $\phi$ (since $\phi \geq 0$ and $\phi(\mathbf{0}) = 0$). Since both $\phi$ and $\kappa_i$ are closed, $H$ is closed. Also by assumption $\exists \mathbf{x}$ such that $A\mathbf{x} \in \mathrm{ri}\,\mathrm{dom}\,H$. Therefore we can apply [19, Theorem 16.3] to conclude that $h^* = (HA)^* = A^* H^*$, where $A^*$ is the adjoint of $A$. Expanding the last expression we get (62).

The second claim follows from the relations

$$\kappa^* = \delta(\kappa^\circ \leq 1) \tag{64}$$
$$\kappa^\circ = \delta^*(\kappa \leq 1), \tag{65}$$

where $\kappa$ is any gauge and $\delta(\cdot) = 0$ if $\cdot$ is true otherwise $\delta(\cdot) = \infty$. Indeed, when $\phi$ is a gauge, so is $h$, and

$$
\begin{aligned}
h^*(\mathbf{x}) &= \min_{\sum_i \mathbf{x}^i = \mathbf{x}} \phi^+(\kappa_1^\circ(\mathbf{x}^1), \dots, \kappa_n^\circ(\mathbf{x}^n)) \\
&= \min_{\sum_i \mathbf{x}^i = \mathbf{x}} \delta(\phi^\circ(\kappa_1^\circ(\mathbf{x}^1), \dots, \kappa_n^\circ(\mathbf{x}^n) \leq 1) \\
&= \delta\left(\left[\min_{\sum_i \mathbf{x}^i = \mathbf{x}} \phi^\circ(\kappa_1^\circ(\mathbf{x}^1), \dots, \kappa_n^\circ(\mathbf{x}^n)\right] \leq 1\right) \\
&= \delta(h^\circ(\mathbf{x}) \leq 1),
\end{aligned}
$$

due to (64). Since both functions (inside $\delta$) are positively homogeneous, we must have (63). ∎

## I  Proof of Proposition 2

Since the polar $\Omega^\circ$ is closed, we have

$$0 = \min_{\boldsymbol{\theta}:\Omega^\circ(\boldsymbol{\theta}) \leq \zeta} \tfrac{1}{2}\|\boldsymbol{\theta} - \mathbf{g}\|_2^2$$

if and only if $\Omega^\circ(\mathbf{g}) \leq \zeta$, therefore

$$\Omega^\circ(\mathbf{g}) = \inf\left\{\zeta \geq 0 : 0 = \min_{\boldsymbol{\theta}:\Omega^\circ(\boldsymbol{\theta}) \leq \zeta} \tfrac{1}{2}\|\boldsymbol{\theta} - \mathbf{g}\|_2^2\right\}. \tag{66}$$

Recall Moreau's identity [19, Theorem 31.5], that is,

$$\mathsf{Prox}_f(\mathbf{g}) + \mathsf{Prox}_{f^*}(\mathbf{g}) = \tfrac{1}{2}\|\mathbf{g}\|_2^2, \tag{67}$$

where $f^*$ denotes the Fenchel conjugate of $f$. Setting $f(\mathbf{g}) = \delta(\Omega^\circ(\mathbf{g}) \leq \zeta)$ we obtain $f^*(\mathbf{g}) = \zeta\Omega(\mathbf{g})$, hence

$$\min_{\boldsymbol{\theta}:\Omega^\circ(\boldsymbol{\theta}) \leq \zeta} \tfrac{1}{2}\|\boldsymbol{\theta} - \mathbf{g}\|_2^2 = \mathsf{Prox}_f(\mathbf{g}) = \tfrac{1}{2}\|\mathbf{g}\|_2^2 - \mathsf{Prox}_{f^*}(\mathbf{g}),$$

which plugged into (66) completes the proof of Proposition 2.

## J  Proof of Proposition 3

The proof is quite straightforward. Let

$$\mathbf{u} := \arg\min_{\boldsymbol{\theta}} \tfrac{1}{2}\|\mathbf{w} - \boldsymbol{\theta}\|_2^2 + \|\boldsymbol{\theta}\|_{\mathsf{TV}} \tag{68}$$
$$\mathbf{v} := \arg\min_{\boldsymbol{\theta}} \tfrac{1}{2}\|\mathbf{u} - \boldsymbol{\theta}\|_2^2 + \|\boldsymbol{\theta}\|_p \tag{69}$$
$$\mathbf{z} := \arg\min_{\boldsymbol{\theta}} \tfrac{1}{2}\|\mathbf{w} - \boldsymbol{\theta}\|_2^2 + \|\boldsymbol{\theta}\|_{\mathsf{TV}} + \|\boldsymbol{\theta}\|_p, \tag{70}$$

then Proposition 3 amounts to claiming that $\mathbf{z} = \mathbf{v}$.

Indeed, by the first order optimality conditions for convex programming [19], we must have

$$\mathbf{0} \in \mathbf{u} - \mathbf{w} + \partial\|\mathbf{u}\|_{\mathsf{TV}} \tag{71}$$
$$\mathbf{0} \in \mathbf{v} - \mathbf{u} + \partial\|\mathbf{v}\|_p, \tag{72}$$

where $\partial\|\mathbf{x}\|$ denotes the subdifferential of the norm $\|\cdot\|$ at point $\mathbf{x}$. It is easy to argue from (72) that $u_i \geq u_j \implies v_i \geq v_j$, therefore exploiting the special structure of $\|\cdot\|_{\mathsf{TV}}$ we can conclude that $\partial\|\mathbf{u}\|_{\mathsf{TV}} \subseteq \partial\|\mathbf{v}\|_{\mathsf{TV}}$. Adding (71) and (72) we obtain

$$\mathbf{0} \in \mathbf{v} - \mathbf{w} + \partial\|\mathbf{v}\|_p + \partial\|\mathbf{v}\|_{\mathsf{TV}}, \tag{73}$$

which implies that $\mathbf{v}$ minimizes (70). Thus $\mathbf{v} = \mathbf{z}$, since both are optimal while the minimizer is unique.

**Algorithm 3** Exact algorithm for the proximal map (74).

---
1: $h_1(-1) = w_1 - 1, h_1(1) = w_1 + 1. \; \mathbb{K}_1 \leftarrow \{(-1, h_1(-1)); (1, h_1(1))\}.$
2: **for** $j = 2, \ldots, m - 1$ **do**
3:     $h_j(z) = z + w_j - \text{Median}(-1, 1, (h_{j-1} + I)^{-1}(w_j + z))$ for $z \in \{-1, 1\}$.
4:     $\mathbb{K}_j \leftarrow \{(-1, h_j(-1)), (1, h_j(1))\}.$
5:     **for all** $(\alpha_i, \beta_i) \in \mathbb{K}_{j-1}$ **do**
6:       **if** $-1 < \alpha_i' := \alpha_i + \beta_i - w_j < 1$ **then**
7:         $\mathbb{K}_j \leftarrow \mathbb{K}_j \cup \{(\alpha_i', \beta_i)\}$
8:       **end if**
9:     **end for**
10: **end for**

---

# K  Fused Lasso: An Efficient Exact Algorithm for Computing $\mathsf{Prox}_{\|\cdot\|_{\mathsf{TV}}}$

Given a vector $\mathbf{w}$, the problem of computing $\mathsf{Prox}_{\|\cdot\|_{\mathsf{TV}}}(\mathbf{w})$ amounts to solving

$$\min_{\boldsymbol{\theta}} \tfrac{1}{2}\|\mathbf{w} - \boldsymbol{\theta}\|_2^2 + \|\boldsymbol{\theta}\|_{\mathsf{TV}}. \tag{74}$$

Applying Moreau's identity [19, Theorem 31.5] we see that $\boldsymbol{\theta}$ minimizes (74) iff for some $\mathbf{z} \in \mathbb{R}^{m-1}$ that solves

$$\min_{\mathbf{z} \in [-1,1]^{m-1}} (z_1 + w_1)^2 + (z_{m-1} - w_m)^2 + \sum_j (z_j - z_{j-1} + w_j)^2, \tag{75}$$

we have $\theta_1 = w_1 + z_1$, $\theta_m = w_m - z_{m-1}$, and $\theta_j = w_j + z_j - z_{j-1}$ for all $2 \le j \le m - 1$.

For $z \in [-1, 1]$, define $H_1(z) = \frac{1}{2}(z + w_1)^2$ and recursively for $2 \le j \le m - 1$ define

$$H_j(z) = \min_{|z_{j-1}| \le 1} H_{j-1}(z_{j-1}) + \tfrac{1}{2}(z - z_{j-1} + w_j)^2. \tag{76}$$

It is readily verified that solving (75) amounts to minimizing $H_{m-1}(z) + \frac{1}{2}(z - w_m)^2$. Inductively, we infer that $H_j$ is a convex piecewise quadratic univariate function. Therefore its derivative, denoted as $h_j$, is increasing and piecewise linear. Denote subdifferential $\partial h_j(1) = [\lim_{z \uparrow 1} h_j(z), \infty)$ and $\partial h_j(-1) = (-\infty, \lim_{z \downarrow -1} h_j(z)]$. Moreover, for all $2 \le j \le m - 1$

$$h_j(z_j) = z_j + w_j - z_{j-1}, \tag{77}$$

$$\text{where} \quad z_{j-1} = \arg \min_{-1 \le z \le 1} H_{j-1}(z) + \tfrac{1}{2}(z_j - z + w_j)^2 \tag{78}$$

$$= \text{Median}(-1, 1, (h_{j-1} + I)^{-1}(z_j + w_j)). \tag{79}$$

Therefore if $h_{j-1}$ has $k$ (linear) pieces, $h_j$ has at most $k + 1$ (linear) pieces (taking into account the end points $z = \pm 1$). Using dynamic programming we can recursively identify all the "kink points" of $h_j$ (denoted as $\mathbb{K}_j$) for $j = 1, \ldots, m - 1$, and hence easily find the minimizer of $H_{m-1}(z) + \frac{1}{2}(z - w_m)^2$, that is, (74).

Thus we can summarize the procedure in Algorithm 3.

Note the space cost is $O(m)$ and upon completion of Algorithm 3, we only have $\mathbb{K}_{m-1}$, based on which the optimal $z_{m-1}^*$ can be found. To recover the optimal $z_1^*, \ldots, z_{m-2}^*$, we backtrack the values of $z_j^*$ and $h_j(z_j^*)$. By (77), it is obvious that for $2 \le j \le m - 1$

$$z_{j-1}^* = z_j^* + w_j - h_j(z_j^*). \tag{80}$$

Then by (79), we have three cases:

- $z_{j-1}^* = -1 \Rightarrow h_{j-1}(z_{j-1}^*) = h_{j-1}(-1)$ which we have recorded in Algorithm 3.
- $z_{j-1}^* = 1 \Rightarrow h_{j-1}(z_{j-1}^*) = h_{j-1}(1)$ which we have also recorded in Algorithm 3.
- $z_{j-1}^* = (h_{j-1} + I)^{-1}(z_j^* + w_j) \Rightarrow h_{j-1}(z_{j-1}^*) = z_j^* + w_j - z_{j-1}^* = h_j(z_j^*)$.

### K.1 More Experiments on Fused Lasso with Comparison to Liu et. al. [26]

We compared two algorithms that solve the proximal operator $\text{Prox}_{\|\cdot\|_{\text{TV}}}$ in fused lasso. One is our dynamic programming (DP) Algorithm 3, and one is from Liu et. al. [26] whose implementation was extracted from the SLEP package[2]. In particular, we randomly generated an $m$-dimensional vector $\mathbf{w}$ and used the two methods to solve

$$\min_{\boldsymbol{\theta}} \tfrac{1}{2}\|\mathbf{w} - \boldsymbol{\theta}\|_2^2 + \lambda\|\boldsymbol{\theta}\|_{\text{TV}}. \tag{81}$$

The components of $\mathbf{w}$ were drawn independently from unit Gaussians, and the dimension $m$ ranged from $10^4$ to $10^6$. We varied $\lambda \in \{0.01, 0.1, 1, 10, 100\}$ and the resulting run time is shown in Figure 4 to 8 respectively. For each combination of $m$ and $\lambda$, 50 random samples of $\mathbf{w}$ were drawn which allowed us to plot the error bar.

It is clear that the run time of both algorithms is linear in $m$. However, our DP algorithm is 2 to 6 times faster than [26], and the margin grows wider as the values of $\lambda$ increase.

Figure 9 shows the total number of kinks generated along the execution of our DP algorithm. It is also linear in $m$ and the slope is 2 to 12 depending on $\lambda$.

Figure 4: Running time (in seconds) of our DP algorithm vs [26] for $\lambda = 0.01$.

Figure 5: Running time (in seconds) of our DP algorithm vs [26] for $\lambda = 0.1$.

Figure 6: Running time (in seconds) of our DP algorithm vs [26] for $\lambda = 1$.

Figure 7: Running time (in seconds) of our DP algorithm vs [26] for $\lambda = 10$.

Figure 8: Running time (in seconds) of our DP algorithm vs [26] for $\lambda = 100$.

Figure 9: Number of pieces in our DP

## Footnotes

[1]This assumption allows us to interpret $\phi(\infty, \ldots)$ as $\infty$.

[2]http://www.public.asu.edu/~jye02/Software/SLEP/index.htm