[Reviews · NeurIPS 2013]

Submitted by Assigned_Reviewer_4

Comments:

This paper proposes a new method for structured sparse learning based on the generalized conditional gradient (GCG) framework. Several polar operators are proposed for different structured sparsity patterns, e.g. group sparsity, path coding. The comparisons with the proposed method with existing ones in the experiments seem promising.

The proposed algorithm is very interesting and seems useful for several applications. There are two major concerns. Theoretically, the GCG has a slower theoretical convergence rate than APG. With the introduced local optimization (Line 5), it is claimed to have faster convergence in practice without solid theoretical supports. As for the experimental comparisons, overall, the experiments are rather illustrative than actual in this paper. The proposed method is compared with standard solvers only on special applications. It is hard to conclude that the proposed algorithm can be considered as serious alternatives for those of structured sparsity problems. The state-of-the-art algorithm on such applications should be included.

Some additional comments:

1. In line 125, it said that “Although GCG has a slower theoretical convergence rate than APG, the introduction of local optimization (Line 5) often yields faster convergence in practice”. What is the support for this conclusion? This paper will be significantly enhanced if there exist solid theoretical supports.

2. In Algorithm 1, Line 4 finds the optimal step sizes for combining the current iterate wk with the direction vk and Line 5 locally improves the objective (1) by maintaining the same support patterns but re-optimizing the parameters. Both of them require accessing the loss function f(x). If the loss function is complex, the computational costs can not be ignored. On the other hand, it is not necessary to access the loss function f(x) in each iteration for the APG. How to Can you make a discussion on this?

3. In section 5.3, the proposed algorithm has been validated in the problem of latent fused lasso. However, it is not compared with the state-of-art algorithms for this problem. For example, J. Liu, et al. An efficient algorithm for a class of fused lasso problems. In KDD2010. Without fair comparison with the state-of-art algorithm, it is hard to say the benefit of the proposed algorithm.
Summary: This paper proposes a new method for structured sparse learning based on the generalized conditional gradient (GCG) framework. However, there are two major concerns for current version of this paper.

Submitted by Assigned_Reviewer_5

The authors build their work on top of the generalized conditional gradient (GCG) method for sparse optimization. In particular, GCG methods require computation of the polar operator for the sparse regularization function (an example is the dual norm if the regularization function is an atomic norm). In this work, the authors identify a class of regularization functions, which are based on an underlying subset cost function. The key idea is a to 'lift' the regularizer into a higher dimensional space together with some constraints in the higher-dimensional space, where it has the property of 'marginalized modularity' allowing it to be reformulated as a linear program. Finally, the approach is generalized to general proximal objectives. The results demonstrate that the method is able to achieve better objective values in much less CPU time when compared with another polar operator method and accelerated proximal gradient (APG) on group Lasso and path coding problems.

While I am not an expert in this field nor did I understand many of the technical details, I did find that the paper was not written clearly and used an unnecessarily large amount of notation. In addition, the motivation is slightly unclear from the start - on one hand an efficient method is desired while on the other hand APG has significantly better convergence guarantees (1/\epsilon vs. 1/\sqrt{\epsilon}) than this GPG-based method. While the results show a very large efficiency improvement, it is unclear to me whether this generalizes outside of these two cases. The lifting construction and reformulation as LP are interesting in their own right but I do not feel sufficiently informed to evaluate them.
Summary: The authors propose (1) a class of regularizers inducing structured sparsity and (2) an efficient optimization method for these regularizers which shows impressive performance improvements over other existing methods. However, theoretical guarantees on this method do not match that of existing methods and it is unclear whether these performance improvements generalize to other data sets.

Submitted by Assigned_Reviewer_6

-- first-round comments --

This paper proposes new methods to compute the polar operator for a class of regularizers based on marginalized modular functions such as group sparsity.

Pros:
- well-written, and to-the-point. Logically organized
- subject is topical (group sparsity, sparsity on DAGS)
- Polar operators have not been studied too much (with respect to practical use) compared to proximal operators. The proposed method seems to be quite good
- Recent interest in conditional gradient/Frank Wolfe means that polar operators are of interest
- Nice secondary results (e.g., prox of lp + TV)

Cons:
- Prop. 2 allows one to compute the polar via the prox, due to the link between the dual functions and polar functions (and between a function and its conjugate, via Moreau's Thm, eq (43)). By itself, the proposition is not too surprising, and I strongly suspect that a "converse" can also be shown (i.e., one can compute the prox via the polar). If this is true, then does the advantage of the GCG over APG still hold? In the experiments, the proxes used in APG were very slow, but couldn't these be sped up to within a factor of 5 of the time to compute the polar? (since going the other way, it took about 5 evaluations of the prox to compute the polar). Also, it would be nice to say something more about the 1D line search; e.g., can one use Newton's method, as in SPGL1 (Friedlander and van den Berg)?

- Actually, is it not possible to find the polar via the prox in another manner? The polar operator for a set S at the point "d" is polar(d)=argmin_{x in S} -d^T x. The prox operator at the point "y" is prox(y)=argmin_{x in S} .5||x-y||^2. Then by expanding the quadratic, we roughly get lim_{alpha -> infinity} prox( -alpha d ) = polar(d). Computationally there are still issues, but it may be an interesting alternative.

- Regarding the exact solution of 1D TV, improving on that in [25]. I do not think this is new; please check recent literature like L. Condat, "A direct algorithm for 1D total variation denoising," preprint hal-00675043, 2011 (which also mentions other exact, direct algorithms, such as the taut string algorithm).

- complexity results used Nesterov's smoothing method, but experiments use L-BFGS. Then, line 362, the paper says "...the method proposed in this paper is not only faster... in practice, it also possesses a strictly better worst case time complexity..." The practical results use L-BFGS, and the worst case time complexity use Nesterov's APG, so this statement is not correct unless you have a complexity bound for L-BFGS.

- The appendix is rather long and contains new propositions. I prefer appendices to be reserved for small details and lemma. If such a long appendix is necessary, is NIPS really the right venue?

Neither a "pro" nor a "con", this paper will likely be hard to read for most readers, since it combines combinatorial-like functions (modularity), convex optimization (polars and duals), and statistics/machine learning (lasso, etc.)

In my opinion, the pros outweigh the cons considerably. I think this is a significant paper.

------------------------
new comments after reading feedback:

- In response to the other reviewer, you wrote "APG also needs to evaluate the loss function if the Lipschitz constant requires adaptive estimation". Actually, it is possible to do a check using just the iterates and the gradients since if the gradient is Lipschitz it is also firmly non-expansive (when scaled by the Lipschitz constant), and so you can use the firmly non-expansive bound to check (see the old Eckstein-Bertsekas paper on the Douglas-Rachford iteration).

Also, this comment
"- complexity results used Nesterov's smoothing method..."
was not addressed. You are claiming to be better in theory and practice, but your theory uses APG and your practice uses L-BFGS, so theory and practice do not mach.
Summary: Novel paper that considers a relevant but not much explored topic: polar operators. Since not many methods have been proposed for these outside the trivial cases, the method in this paper is a significant improvement.
Author Feedback

Author rebuttal: We thank the reviewers for their valuable comments.

====
Reviewer_4:

Q1: GCG has slower theoretical rate but converges faster in practice.

A: True, the current analysis indicates that GCG in theory converges slower than APG. However, the former possesses some special properties that are not shared by APG:

1. We get to control the sparsity of intermediate iterates in GCG, because each iteration adds one "atom".

2. GCG naturally maintains an active set of variables that can be locally optimized. As observed by many groups (on a variety of problems) such as [14, 15], local optimization over an active set can significantly accelerate convergence. Indeed, Liu et al. [25] also used the active set technique (via local optimization and gradient descent) when computing the proximal map for the fused lasso, and obtained significant acceleration over APG. However, different from Liu et al. who applied the active set technique in the inner loop (i.e. the proximal operator), we combine GCG with local optimization in the outer loop.

3. Incorporating local optimization in the *outer* loop of APG is challenging, because APG maintains two sequences of iterates. Locally optimizing one sequence will break the synergy with the other.

Therefore, GCG demonstrates novel promise and is worth studying, despite the slower rate in its vanilla form. The main goal of this paper is not to replace APG with GCG, but to point out that in some cases, by carefully leveraging the special properties of GCG, it can be made very competitive. Considering the great interest that conditional gradient has received in many communities like machine learning, and the relatively little work on computing the polar operator, we believe our results can help understand the relative merits of GCG and APG.


Q2: Comparison with Liu et al.

A: In this experiment on fused lasso, we applied our Algorithm 2 (Appendix I) to solve the proximal map that is used by both APG and GCG. During the rebuttal period, we complemented our Figure 3 by trying APG that uses the proximal algorithm in Liu et al. (called APG-Liu). We observed that in APG, our proximal map allows the overall optimization to converge with slightly less CPU time than APG-Liu. But GCG further reduces the time by 50%.

Also during the rebuttal period, we further conducted a direct comparison between our proximal algorithm and Liu et al. (SLEP 4.1). Both methods were invoked with an m-dimensional vector, whose components were drawn independently from unit Gaussians (m = 10^4 to 10^6). We varied \lambda in {0.01, 0.1, ..., 100}. The computational cost of our algorithm exhibits clear linearity in m with an even slope. It is 2 to 6 times faster than Liu et al., with the margin growing wider as the values of \lambda increase. See detailed plots in above link.


Q3: The proposed method is compared with standard solvers only on special applications.... The state-of-the-art algorithm on such applications should be included.

A: We did compare with the state-of-the-art algorithms for CUR-like matrix factorization and path coding, which are typical sparse coding problems. In fact, as we stated in the paper, the implementation of APG was exactly the SPAMS package. They did not show up in Figure 1 and 2 because the time cost for a single call of the proximal update in APG was enough for GCG to converge to a final solution. Other packages such as SLEP could be tested on. For a rebuttal on fused lasso please see Q2.


Q4: Local optimization requires the access of the function value. If the loss function is complex, the computational costs cannot be ignored. On the other hand, it is not necessary to access the loss function f(x) in each iteration for the APG.

A: All first order algorithms, including APG and GCG, require access to the gradient of the loss function. APG also needs to evaluate the loss function if the Lipschitz constant requires adaptive estimation. Besides, in most applications, the function value is no more expensive than evaluating the gradient.

In practice, our overall scheme indeed requires fewer function/gradient evaluation than APG. This is because: 1. local optimization can substantially reduce the number of iterations required by GCG. 2. Local optimization, which itself is based on quasi-Newton methods, empirically converges in a small number of iterations, thanks to the small size of support and the low accuracy required.


====
Reviewer_5:

Q1: GCG has a slower rate than APG

A: Please see Q1 for Reviewer_4.


====
Reviewer_6:

Q1: Converse of polar via prox operator.

A: Indeed, it is possible to reduce the prox to a sequence of polar evaluations, for instance one can just apply the GCG algorithm to the prox. However, for all reductions that we are aware of, it is usually less efficient to reduce prox to polar than the other way around.

Q2: Can the prox be sped up?

A: In the first part of the paper (Section 3), we do not reduce the polar to prox but compute polar directly through a sequence of developments. So far it is not clear whether or not one can further improve the prox or polar in these examples (either would be a nice contribution). Also, we do not claim that polar is always cheaper than prox or the converse. It certainly depends on the regularizer. What we try to say is that the polar, if not cheaper, is at most a couple of proxs (using the reduction). (Note that l_inf norm costs O(n), not O(n log n))

Q3: 1D line search

A: One can certainly use the Newton method, although we haven't tried it yet.

Q4: Other way to reduce polar operator to prox operator.

A: Very interesting proposal, we will try it out.

Q5: 1D total variation.

A: Thanks for pointing out the reference, which is also based on dynamic programming. Our derivation is completely different, leading to a rather different algorithm. We will cite the reference.